# Understanding LoRA As Knowledge Memory: An Empirical Analysis

## Abstract

Continuous knowledge updating in Large Language Models (LLMs) is a critical challenge. While existing methods like Retrieval-Augmented Generation (RAG) and In-Context Learning (ICL) offer solutions, they are constrained by retrieval quality and context length. Departing from the conventional view of LLM memory that relies on context, this work highlights a novel parametric approach via Low-Rank Adaptation (LoRA). Although a few studies have hinted at this potential, LoRA's mechanics and optimal usage as a memory component remain largely unexplored. To bridge this gap, we conduct the first systematic and comprehensive empirical study of LoRA-based knowledge memory. Our analysis spans multiple dimensions, including the fundamental memory characteristics of LoRA, how to optimize a single LoRA, the possibilities of combining multiple LoRAs, and its synergy with existing methods in complex scenarios. Ultimately, this paper presents the first systematic framework for LoRA-based memory, offering foundational insights and actionable guidelines to future research and application.[1]

## 1 Introduction

Large Language Models (LLMs) have demonstrated unprecedented capabilities in natural language understanding and generation, catalyzing innovation across a multitude of domains (Romera-Paredes et al., 2024; Chen et al., 2021; Tu et al., 2024; Wei et al., 2022). However, a fundamental limitation persists: their reliance on the static knowledge embedded within their parameters at the time of pre-training. Consequently, the ability to seamlessly integrate information or to continuously learn and retain new, domain- or user-specific knowledge remains a challenge.

Several mainstream approaches have been developed to update or supplement the knowledge of LLMs. Full model retraining is often computationally prohibitive for frequent use. In-Context Learning (ICL) (Brown et al., 2020) offers an alternative by supplying information directly within the prompt, but this approach is constrained by the limited context window size and a quadratically scaling inference cost. Another popular paradigm, Retrieval-Augmented Generation (RAG) (Lewis et al., 2020), augments the context by retrieving real-time information. However, its performance depends heavily on retrieval efficacy, its understanding can be fragmented by a reliance on top-k passages, and it is similarly subject to context window constraints (Weller et al., 2025).

Low-Rank Adaptation (LoRA) (Hu et al., 2022) has become a Parameter-Efficient Fine-Tuning (PEFT) (Mangrulkar et al., 2022) method for mitigating the prohibitive costs of fine-tuning LLMs. While its primary application has been for task adaptation, its properties of efficiency and modularity motivate an alternative application: its use as a dedicated module for knowledge storage.

Recent studies have begun to explore this direction. PRAG (Su et al., 2025) proposes a Parametric RAG system that retrieves and merges document-specific LoRAs at inference. SEAL (Zweiger et al., 2025) introduces a meta-learning framework to generate optimal synthetic data, while PnP (Caccia et al., 2025) employs Deep Context Distillation to align a LoRA module with a teacher model. Despite their contributions, several aspects remain to be addressed. PRAG and SEAL are primarily evaluated on memorizing relatively short contexts. Moreover, the complex outer-loop processes or teacher distillations in SEAL and PnP incur computational overhead. These methods are also largely

---

[1]All prompts used in our experiments are provided in the Appendix for reproducibility. Code will be made publicly available soon.

confined to optimizing a single LoRA module, leaving the dynamics and scalability of multi-module systems less explored. While PRAG implements multi-LoRA merging, the fundamental analysis of merging methodologies or the optimal number of modules to combine remains underexplored.

To bridge this gap, we conduct the first systematic and comprehensive empirical study of LoRA-based knowledge memory. Unlike prior works that propose specific systems, we aim to establish the fundamental physics of LoRA memory. Our analysis spans multiple dimensions, including quantifying the finite storage capacity and saturation points based on rank, proving that lower ranks are surprisingly more parameter-efficient. We further investigate optimal data formats, demonstrating how synthetic QA and Summaries outperform raw text. Crucially, we analyze multi-module systems, identifying parameter interference as a key bottleneck and empirically confirming that advanced merging algorithms like TIES significantly outperform the linear methods used in prior work. Ultimately, this paper presents the first systematic framework for LoRA-based memory, offering foundational insights and actionable guidelines to future research and application.

## 2 RELATED WORK

**LoRA.** Originally a fine-tuning method for task adaptation (Hu et al., 2022), LoRA is widely used for modular continual learning (Ostapenko et al., 2024; Li et al., 2025), (Pletenev et al., 2025; Liang et al., 2025). Recent works repurpose LoRA as "knowledge memory" (Caccia et al., 2025; Zweiger et al., 2025) making it a fundamental technique for parameter-efficient adaptation across various domains, although often with computational overhead.

**LLM Memory.** Extending LLM memory beyond temporary, non-parametric methods like In-Context Learning (ICL) (Brown et al., 2020) is a central challenge. Solutions range from non-parametric external memory systems (Xu et al., 2025; Chhikara et al., 2025) to parametric approaches, which include layer duplication (Wang et al., 2024) and long-context architectural adaptations (Behrouz et al., 2024; 2025; Dao & Gu, 2024).

**RAG.** RAG (Lewis et al., 2020) externalizes knowledge by retrieving documents, but its effectiveness is often constrained by the representational limits of embedding-based retrieval (Weller et al., 2025) and the narrow context provided by top-k selection (Kuratov et al., 2024).

**Multi-LoRA and Parameter-Space Interpolation.** An emerging area involves composing multiple LoRA modules via parameter-space interpolation (Huang et al., 2023; Feng et al., 2024; Dou et al., 2024; Prabhakar et al., 2025) or scalable routing mechanisms (Fleshman & Van Durme, 2025). While frameworks like Parametric RAG (Su et al., 2025), (Tan et al., 2025) merge modules for retrieval systems, we distinguish our work by providing a foundational analysis of LoRA's intrinsic storage properties in long-context scenarios.

**Synthetic Data.** The structure of fine-tuning data is critical; transforming raw text into synthetic formats has been shown to build robust knowledge representations in LLMs (Lampinen et al., 2025; Park et al., 2025; Lin et al., 2025; Yang et al., 2025b; Zhang et al., 2024).

## 3 CHARACTERIZING THE MEMORY ABILITY OF LORA

Our initial investigation aims to uncover fundamental properties of a single LoRA module as a memory unit. Is LoRA's capacity for memorizing new knowledge scalable, and what are its practical limits? To answer this question, we select two datasets: PhoneBook (PB; see Appendix B) and existing CounterFact (CF) (Meng et al., 2022).

PB is a synthetic dataset consisting of symbolic associations, specifically pairs of fictional names and phone numbers, which are unrelated to the model's pre-trained knowledge. CF introduces counterfactual statements such as "Paris is in Italy", which directly contradict the model's pre-trained beliefs. Both datasets are programmatically scalable, allowing us to precisely control the amount of knowledge in tokens and observe scaling trends.

For evaluation, we use exact match for PB and efficacy score for CF. Our analysis in this section is based on a grid search over LoRA ranks (2 to 1024) and amount of knowledge (1k to 20k tokens) using Llama-3.1-8B and Llama 3 tokenizer (Grattafiori et al., 2024). Full experiment details are in Appendix C.

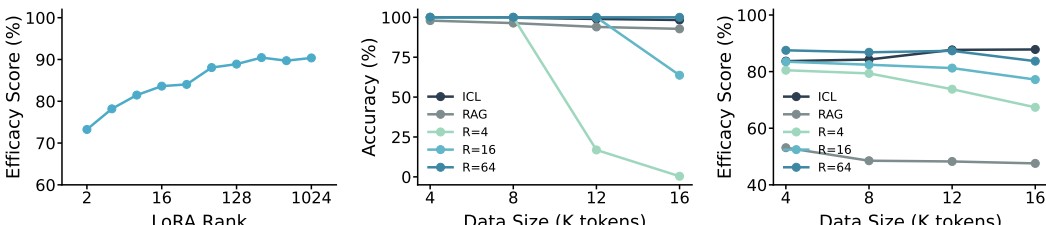

Figure 1: **(Left)** Performance trend on the CounterFact-10k dataset as Rank increases. **(Center)** Performance on PhoneBook for various ranks as data length increases. **(Right)** Performance on CounterFact for various ranks as data size increases.

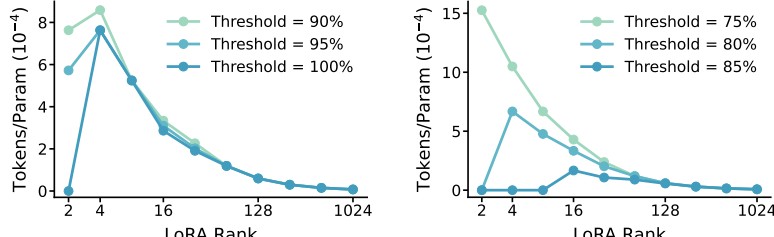

Figure 2: Efficiency for the two datasets. **(Left)** PhoneBook results. **(Right)** CounterFact results.

## Q1. Is Memory Capacity Scalable with LoRA Rank?

Our core thesis posits LoRA as a scalable knowledge memory, with rank serving as the key hyper-parameter governing the trade-off between capacity and cost. A prerequisite is a positive correlation between rank (i.e., parameter count) and storage capacity, which our results confirm: across both PhoneBook and CounterFact, higher ranks enabled greater memorization. Figure 1 (left) illustrates this on CF-10K, where efficacy steadily increases with rank, indicating that larger parameter budgets allow LoRA modules to encode more complex knowledge (see Appendix D for full details).

> **Implication:** LoRA's knowledge capacity is scalable. This validates its use as a controllable memory module, transcending its role as a mere fine-tuning method.

## Q2. Is There a Finite Capacity Limit? How Is It Determined?

We next examine LoRA's capacity limit for memorization. Results reveal a rank-dependent *saturation* effect: at fixed rank, performance declines as knowledge load increases, with lower ranks saturating earlier. As shown in Figure 1 (center/right), higher-rank modules, by contrast, sustain performance, confirming larger ranks yield higher capacity ceilings (details in Appendix E).

> **Implication:** A LoRA module has a finite capacity is determined by its rank.

## Q3. Is Using The Highest Rank Always The Most Efficient Choice?

While our findings in Q1 and Q2 establish that higher ranks yield greater absolute capacity, a critical engineering question remains: Is large rank always better? In practice, high ranks incur substantial costs in parameters, training time, and memory usage. This motivates an analysis of *parameter efficiency*—the amount of information stored per parameter. We investigate whether capacity scales linearly with rank or if it exhibits diminishing returns. To quantify efficiency, we determine the maximum amount of knowledge a LoRA can absorb before becoming "saturated" and normalize by its parameter count:

$$\text{Efficiency} = \text{Max tokens memorized above threshold}/\text{Number of parameters} \qquad (1)$$

We find that the highest rank is not the most parameter-efficient. Figure 2 shows that efficiency curves are non-monotonic, reaching a peak at specific ranks before declining. Notably, both PB and CF achieve maximum efficiency at **low ranks**. This indicates that beyond a certain point, increasing rank yields diminishing returns in capacity relative to the parameter cost (details in Appendix F).

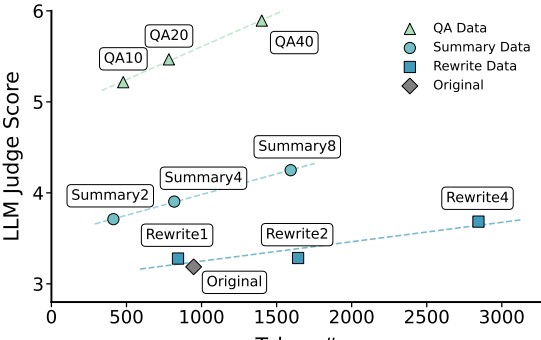

| Methods | Performance |
|---|---|
| Original | 3.187 |
| QA | 5.893 |
| Original + QA | 6.300 |
| Summary + QA | 6.380 |
| Rewrite + QA | 6.650 |
| Original + Summary + Rewrite + QA | **6.822** |

Figure 3: **(Left)** Performance scaling with different synthetic data augmentation methods showing QA, Summary, and Rewrite approaches. **(Right)** Performance when combining multiple augmentation methods, demonstrating complementary effects.

> **Implication:** A trade-off exists between absolute memory capacity and parameter efficiency in LoRA modules. Simply maximizing rank can lead to resource waste.

**Discussion.** First, we confirmed that its capacity is **scalable with rank**, validating its viability as a memory module. Second, we showed that capacity is **finite and rank-dependent**, highlighting the challenge of managing a limited memory budget. Third, by introducing **parameter efficiency**, we revealed that lower-rank modules often yield higher efficiency, exposing a trade-off between absolute capacity and resource cost. These findings motivate two directions:

1. The imperative to optimize a **Single LoRA** module. Since a LoRA's capacity is finite, its utilization must be maximized. Using capacity on redundant or low-value information is suboptimal. This necessitates strategies for refining the knowledge before it is stored. In Section 4 we will analyze methodologies for enhancing the knowledge density of a single LoRA.

2. The motivation for **Multi-LoRA** systems. The capacity limit and efficiency of smaller ranks motivate modular architectures. Partitioning knowledge across multiple LoRAs can be efficient, but raises new challenges of query routing and knowledge integration, which we analyze in Section 5.

## 4   OPTIMIZING KNOWLEDGE MEMORIZATION IN A SINGLE LoRA

**The PaperQA Benchmark.** To rigorously evaluate a LoRA module's ability for internalizing new knowledge, we introduce the PaperQA benchmark, which has two key distinctions from existing benchmarks. First, constructed from very recent academic papers (NeurIPS '24, ICLR '25 and ICML '25), it minimizes the knowledge leakage risk. Second, it facilitates a multi-faceted evaluation of a single document's comprehension by generating a diverse set of questions for each paper. These questions assess three key cognitive dimensions: information recall, contextual comprehension, and logical inference. Answers are evaluated via an LLM-as-judge on a 0-10 scale, capturing a nuanced depth of understanding beyond simple accuracy. Details for the dataset are provided in Appendix G.

**Q4. How Does Synthetic Data Enhance Single LoRA's Knowledge Memorization?**
Our analysis confirmed that LoRA capacity is finite, requiring efficient use through data refinement. While prior work shows that synthetic data aids knowledge learning in LLMs (Lampinen et al., 2025; Park et al., 2025), it remains unclear whether it is scalable or how different augmentation strategies compare. To fill this gap, we evaluate three approaches. **QA, Summary, Rewrite** against a raw text baseline, varying data size with GPT-4.1 (Achiam et al., 2023) generation (details in Appendix H).

As shown in Figure 3 (left), all synthetic data types surpassed the raw text baseline. QA performed best due to its close alignment with the evaluation format and also delivered the highest efficiency, showing greater performance gains per token than other methods.

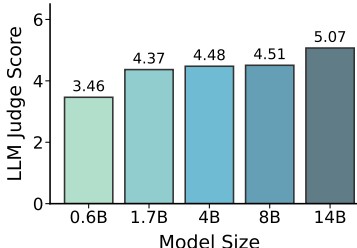 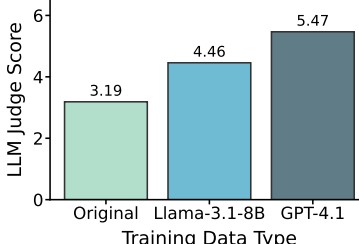

Figure 4: **(Left)** Performance across different Qwen3 model sizes. **(Right)** Performance comparison when using GPT vs. Llama for synthetic data generation.

> **Implication:** To maximize the performance of a single LoRA, it is highly effective to transform raw data into high-density, task-aligned formats, such as QA.

**Q5. What Are the Effects of Combining Synthetic Data Formats?**

Having identified QA as the most efficient single synthetic format, we next investigate whether combining formats can yield synergistic benefits and surpass single-method ceilings. To test this, we construct combined datasets by concatenating QA, Summary, and Rewrite data from the previous experiment. As shown in Figure 3 (right), all combinations outperformed QA alone, confirming the benefits of diversity. While Rewrite lagged behind Summary in isolation, pairing it with QA surpassed the QA+Summary result. The best scores came from the most diverse mixture, showing that complementary signals maximize LoRA's retention (details in Appendix I).

> **Implication:** Combining diverse synthetic data formats yields complementary benefits, enabling LoRA modules to surpass the performance ceilings of single-format training.

**Q6. How Does the Scale of Base Models Affect LoRA Knowledge Internalization?**

We examine how base model scale influences LoRA's knowledge internalization, a key factor for balancing performance and cost. To this end, we fine-tuned LoRA on Qwen-3 (Yang et al., 2025a) models ranging from 0.6B to 14B with identical training data and hyperparameters. Figure 4 (left) shows performance improved with model size but non-linearly: significant gains from 0.6B-1.7B and 8B-14B, minimal improvement from 1.7B-8B. Unlike ICL which leverages base model reasoning, LoRA primarily stores knowledge in its added parameters, making it less sensitive to base model capacity in certain ranges (details in Appendix J).

> **Implication:** Larger base models improve LoRA performance, but non-linearly with diminishing returns in mid-range scales (1.7B-8B).

**Q7. Does Synthetic Data Generator Quality Impact LoRA Performance?**

Following Q4 and Q5's findings on synthetic data benefits, we investigate whether generator model quality affects outcomes—a practical trade-off between costly API models and local alternatives. We compare LoRA trained on QA data from GPT-4.1 versus Llama-3.1-8B. Figure 4 (right) shows GPT-4.1-generated data consistently yielded higher performance than Llama-3.1-8B data. Superior generators produce more accurate, logically sound, and comprehensive training data, with these quality differences directly transferring to final LoRA performance (details in Appendix K).

> **Implication:** Stronger data generators yield better knowledge internalization in LoRA.

## 5 SCALING TO MULTI-LORA SYSTEMS

Section 3 highlighted the inherent capacity limits and efficiency trade-offs of single LoRA modules, motivating the need for multi-module architectures. Prior work such as PRAG (Su et al., 2025) has adopted multiple LoRAs, but without systematic justification or analysis of key design dimensions—routing, merging, and partitioning. We present the first comprehensive empirical analysis of Multi-LoRA systems, demonstrating their potential under ideal conditions, quantifying performance degradation under realistic routing, and evaluating merging strategies to alleviate these losses.

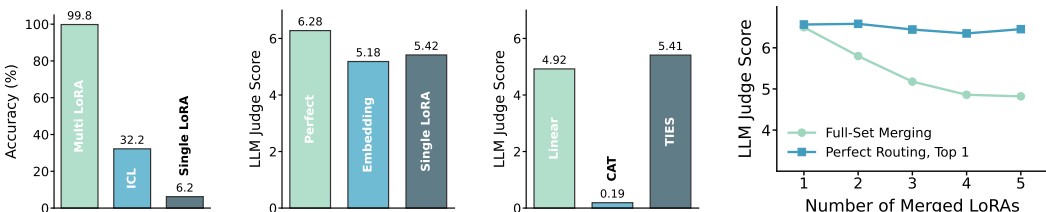

Figure 5: **(First)** Multi-LoRA PoC on 64K PhoneBook. **(Second)** Performance gap between perfect and RAG routing. **(Third)** Merging strategies comparison. **(Fourth)** A comparison between selecting a single optimal LoRA and merging N LoRAs.

## Q8. How Can We Utilize Multiple Small LoRAs to Outperform a Single Large LoRA?

Before addressing the complexities of routing and merging, we first establish a proof of concept for the Multi-LoRA approach. Using the PhoneBook dataset with 64K tokens, we compare three configurations under an identical parameter budget: (1) a full-context ICL baseline, (2) a single large LoRA module, and (3) multiple small LoRA modules, each trained on a partition of the data. For Multi-LoRA, we assume a *perfect routing* oracle that always selects the correct module.

As shown in Figure 5 (first), ICL performance degraded severely with the long context, while a single LoRA lacked the capacity to memorize the dataset. In contrast, Multi-LoRA achieved excellent performance by distributing the knowledge across modules and activating the correct one at test time. Although this relies on the idealized assumption of perfect routing, it demonstrates the feasibility of the Multi-LoRA approach and motivates routing as the next challenge (details in Appendix L).

> **Implication:** Employing multiple small LoRA modules enables greater memory capacity, contingent on accurate routing.

## Q9. How Much Performance Loss Does a More Practical Routing Incur?

Q8 demonstrated the potential of Multi-LoRA under perfect routing, but real-world systems must handle imperfect information. To quantify the performance gap between ideal and realistic scenarios and identify if routing is a critical bottleneck, we designed a new experiment using the PaperQA dataset, as it is better suited for embedding-based routing than the PhoneBook dataset. In this setup, we compare three configurations: (1) Perfect Routing, (2) RAG-based Routing using a standard text-embedding model, and (3) a Single-LoRA baseline.

As show in Figure 5 (second), RAG-based routing suffered a performance drop of about 17.5% relative to the oracle, and even underperformed the Single-LoRA baseline. This highlights how misrouting can be more harmful than no partitioning at all. Recent work (Weller et al., 2025) showed that embedding-based retrieval has fundamental representational limits, making it inherently unable to capture all possible query–document relationships (details in Appendix M).

> **Implication:** The gap between ideal and realistic routing necessitates either improved routing mechanisms or alternative strategies to mitigate routing uncertainty.

## Q10. Can Merging Multiple LoRAs Mitigate Routing Error?

Given the inherent uncertainty in routing, we investigate whether retrieving and merging multiple LoRAs can yield more robust performance than relying on a single retrieved module. We evaluate three representative approaches widely adopted in recent works: (1) **Linear averaging**, a common baseline for parameter merging; (2) **Concatenation (Cat)**, previously used in modular systems such as PRAG (Su et al., 2025); and (3) **TIES**, a parameter-pruning method designed for merging adapters (Yadav et al., 2023).

As shown in Figure 5 (third), TIES not only outperformed RAG-based routing (left) but also matched the Single-LoRA baseline (right), demonstrating its ability to mitigate routing errors. In contrast, Linear merging lagged behind, and Cat failed, confirming that naively concatenating parameters is not a viable strategy (details in Appendix N).

> **Implication:** Sophisticated merging methods like TIES can preserve the knowledge within each LoRA, thereby compensating for suboptimal routing decisions.

| Method | Llama-3.2-1B | Llama-3.1-8B |
|---|---|---|
| *Closed Book* | | |
| Base model | 9.08 | 13.08 |
| KM$_{SDCD}$ | 14.00 | 23.05 |
| Single LoRA | **23.81** | **27.05** |
| Multi-LoRA Top1 | 16.87 | 19.95 |
| Multi-LoRA Top3 | 16.85 | 22.42 |
| *Open Book* | | |
| ICL | 24.52 | 33.81 |
| RAG | 21.90 | 29.20 |
| Single LoRA + ICL | 24.73 | 35.39 |
| Single LoRA + RAG | 24.12 | 32.18 |
| Multi-LoRA Top1 + ICL | 20.57 | 33.62 |
| Multi-LoRA Top1 + RAG | 19.22 | 25.03 |
| Multi-LoRA Top3 + ICL | **26.41** | **38.78** |
| Multi-LoRA Top3 + RAG | 24.53 | 35.55 |

Table 1: ROUGE-L scores on NarrativeQA.

### Q11. How Does the Number of Merged LoRAs Affect Performance?

The process of routing and selecting the top-N relevant LoRA modules for a given query involves a fundamental trade-off. A small N increases the risk of routing failure, where the optimal LoRA is missed. Conversely, a large N increases the likelihood of including relevant information but introduces significant risks of parameter interference and knowledge dilution during merging. A quantitative understanding of this relationship is crucial for optimizing Multi-LoRA systems.

To investigate this trade-off, we quantitatively analyzed the impact of N (the number of merged LoRA modules) on performance, progressively increasing it from 1 to 5. To isolate the effect of interference, we used the simple Linear Merging method for this analysis. Our results, shown in Figure 5 (fourth), revealed that performance monotonically decreased as N increased. This confirms that knowledge dilution and parameter interference accumulate with each additional LoRA, rapidly degrading system performance (details in Appendix O).

> **Implication:** Merging multiple LoRAs presents a trade-off: although merging can synthesize diverse knowledge, parameter interference degrades performance.

## 6  A CASE STUDY ON LONG, COMPLEX DATA

While previous experiments used datasets with clearly segmented corpora (e.g., PhoneBook, PaperQA), real-world scenarios often involve long-form documents without explicit boundaries. Building on the lessons from these controlled settings, we now push our analysis into a more challenging domain with the NarrativeQA (NQA) (Kočiský et al., 2018) benchmark. NQA presents documents spanning tens of thousands of tokens, where many questions demand synthesizing information across multiple paragraphs. This makes it a natural stress test for Multi-LoRA, as each module only sees a partial view of the text, making inter-chunk reasoning particularly difficult. Full details of NQA are provided in Appendix P.

We selected 40 narratives and used Llama-3.1-8B and Llama-3.2-1B as base models. Each document was partitioned into 2K-token chunks, with a LoRA trained per chunk. To mitigate noise from raw segments, summaries were used to guide QA generation. At inference, we employed a RAG-based retriever to select the top-1 and top-3 most relevant modules, which were then merged using TIES, and performance was evaluated with multi-reference ROUGE-L. To benchmark our approach, we compare it against three baselines: full-context in-context learning (ICL), retrieved generation-augmented (RAG), and KM$_{SDCD}$ (Caccia et al., 2025).

### Q12. How Does LoRA Memory Perform on Long-Context Multi-Hop QA Task?

On the NarrativeQA dataset, the Multi-LoRA system notably underperformed the single LoRA (Table 1). While the extended document length is theoretically advantageous for a modular Multi-LoRA approach, this performance gap stems from the compounding losses identified in Q9 and

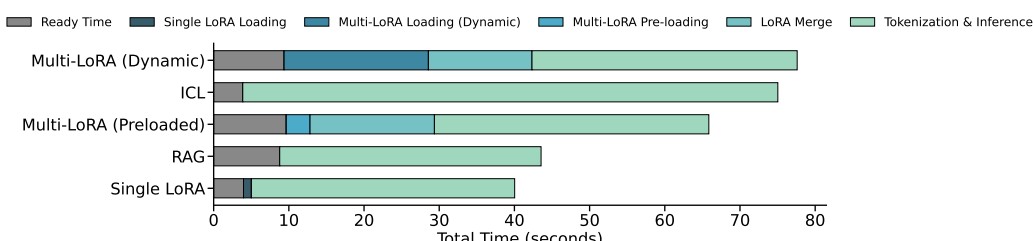

Figure 6: A breakdown of total processing time for different methods. The chart dissects the latency into key stages, including ready time, adapter loading, merging, and inference.

Q10—retrieval failures and merging interference. Furthermore, NarrativeQA's partial multi-hop nature compounds the challenge, as chunk-based approaches struggle when answers require information spanning multiple segments, a well-documented limitation of retrieval-based methods (Kuratov et al., 2024; Weller et al., 2025) (details in Appendix Q).

> **Implication:** For tasks requiring multi-hop reasoning over long documents, the architectural benefits of chunk-based modularity can be undermined by the errors of retrieval and merging.

### Q13. Can LoRA Perform Better When Used With External Context?

Given the limitations of Multi-LoRA's fragmented memory, we test whether supplying external context at inference can help. We build hybrid systems by pairing Single LoRA and Multi-LoRA with ICL or RAG, using either the top-1 or top-3 retrieved modules. These are benchmarked against standalone ICL, RAG, and closed-book LoRA baselines. As shown in Table 1, adding external context consistently enhances performance. The gains are most substantial for Multi-LoRA, where context compensates for the precision loss inherent in merging modules. Notably, ICL yields a greater performance uplift than RAG. This suggests that the continuous, global context supplied by ICL is uniquely effective at restoring the narrative cohesion lost among the fragmented LoRA modules, a weakness that snippet-based retrieval from RAG cannot fully overcome (details in Appendix R).

> **Implication:** LoRA achieves stronger performance when paired with external context, outperforming standalone LoRA, RAG, or ICL. This shows that LoRA is effective as a complementary parametric memory rather than a substitute.

### Q14. Can Merging LoRAs Improve Contextual Continuity?

Beyond relying on external context to maintain continuity in Multi-LoRA systems, we investigate the direct synthesis of knowledge from multiple LoRA modules for multi-hop QA. Our empirical results in Table 1 demonstrate that merging the top-3 modules outperforms selecting the top-1, with the sole exception of the 1B model in the closed-book setting. This advantage is particularly pronounced when augmented with external context from ICL or RAG, indicating that the composition of specialized LoRA memories provides a more robust and comprehensive knowledge foundation than any single module alone (details in Appendix S).

> **Implication:** Merging multiple LoRAs is an possible strategy for improving the model's intrinsic contextual continuity, and this positive effect is amplified when combined with external context.

### Q15. How does LoRA benefit in time?

While previous sections established the performance of LoRA-based memory, practical viability also depends on computational efficiency. Context-based methods like ICL and RAG are computationally expensive, as they must process long context windows for every query. This analysis quantifies whether LoRA, by internalizing knowledge into its parameters, offers a more efficient inference alternative. We benchmarked the time to process 30 sequential questions from a single NarrativeQA document on a Llama-3.1-8B model.

Figure 6 illustrates the results. As hypothesized, LoRA-based methods exhibit shorter pure inference times by eliminating the need to process thousands of context tokens per query. However, they introduce overhead from loading and merging LoRA modules. For Single LoRA, this is a minimal,

one-time cost to load and attach the module. To mitigate the prohibitive I/O latency of dynamically loading modules in the Multi-LoRA setup, we employ a pre-loading strategy where all document-relevant modules are loaded into GPU memory beforehand. This strategy proves highly effective: despite the remaining overhead of merging LoRAs for each query, our Multi-LoRA system achieves a lower total processing time than the ICL-based approach (details in Appendix T).

> **Implication:** In scenarios requiring repeated and interactive access to a consistent knowledge base, the LoRA-based memory approach offers a substantial computational advantage over context-dependent methods.

## 7 CONCLUSION

This paper presents the first systematic investigation into Low-Rank Adaptation (LoRA) as a knowledge memory for Large Language Models. Our findings offer a foundational set of guiding principles for researchers and practitioners aiming to leverage this paradigm, moving from the properties of a single module to the strategic architecture of complex memory systems.

**Treat the Single LoRA as a Resource-Constrained Unit.** Our analysis confirms that a single LoRA module is a finite but scalable memory unit. Its capacity grows with rank, but this comes at a steep cost to efficiency. We found that maximal rank is not the optimal strategy; parameter efficiency peaks at smaller ranks. This establishes a core design principle: large-scale memory systems should be composed of multiple small, efficient modules rather than a single, monolithic one, thereby optimizing the parameter budget.

**Maximize Knowledge Density Through Data Engineering.** Given that LoRA memory is a finite resource, its utilization must be maximized. We demonstrated that storing raw text is highly inefficient. The key to effective knowledge consolidation lies in transforming information intohigh-density, task-aligned synthetic data. A diverse curriculum combining formats like Question-Answering, summaries, and rewrites dramatically improves retention. This principle underscores that the quality of a LoRA memory is fundamentally bounded not just by its architecture, but by the sophistication of the data engineering pipeline that fuels it.

**Architect for Modularity, but Master the System-Level Trade-offs.** Scaling to multi-LoRA systems unlocks vast memory capacity but introduces critical system-level challenges. Our findings pinpoint routing and merging as the primary bottlenecks. While theoretically powerful, a modular system's real-world performance is dictated by the ability to mitigate routing errors and destructive parameter interference during merging. This leads to a crucial insight: for tasks requiring multi-hop reasoning across interconnected information, the compounding errors of a modular system can undermine its benefits. The architectural guideline is to balance the promise of modularity with robust solutions for these new, complex challenges, such as using sophisticated merging algorithms on a small number of retrieved modules.

**Embrace Hybrid Systems for a Synergistic Future.** Perhaps our most critical finding is that LoRA-based memory should not be viewed as a replacement for context-based methods like RAG or ICL, but as their powerful, complementary partner. LoRA offers a highly efficient solution for consolidated, frequently accessed parametric knowledge, providing significant advantages in inference speed and cost. However, peak performance on complex reasoning tasks was achieved in hybrid systems that combine this stable parametric memory with the dynamic, non-parametric context provided by RAG. The ultimate strategy lies in this synergy: leveraging the intelligent integration of parametric and non-parametric memory to build more robust, knowledgeable, and efficient AI systems. This represents a key and promising direction for the future of AI.

## LLM Usage Statement

During the preparation of this work, the authors utilized Large Language Models (LLMs) to assist in various tasks. Specifically, we employed **Cursor** and **Anthropic's Claude** for assistance with code implementation, validation, and debugging. For the manuscript preparation, we used **Google's Gemini** to improve grammar, clarity, and overall readability. The authors reviewed, edited, and take full responsibility for all content, ensuring the scientific integrity and originality of this work.

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

## A    RELATED WORKS IN DETAIL

**LoRA.** Low-Rank Adaptation (LoRA) (Hu et al., 2022) has become one of the most widely adopted parameter-efficient fine-tuning (PEFT) methods. Beyond standard fine-tuning, recent works have explored leveraging LoRA within *continual learning* frameworks, where modular adapters are employed to facilitate incremental updates while mitigating catastrophic forgetting (Ostapenko et al., 2024; Li et al., 2025) (Pletenev et al., 2025; Liang et al., 2025). While these approaches share the goal of task adaptation, more direct attempts to utilize LoRA as a knowledge memory module have recently emerged, such as SEAL (Zweiger et al., 2025) and PnP (Caccia et al., 2025).

Zweiger et al. (2025) proposed SEAL, a meta-learning framework where an LLM learns to generate its own synthetic training data via reinforcement learning. However, SEAL relies on a computationally expensive outer loop to train a generator LLM, often targeting much longer contexts which limits its practical applicability. Similarly, Caccia et al. (2025) introduced PnP, which utilizes a Deep Context Distillation (DCD) objective to align a single LoRA with an in-context teacher model. This method also incurs significant computational overhead due to the distillation framework aligning

hidden states across all layers. While our work shares the exploration of data augmentation strategies for LoRA training with these studies, we distinguish our approach by eliminating the computational overheads associated with SEAL's outer loop and PnP's distillation. Rather than focusing on proposing a specific training method for a *single* LoRA module, we provide a comprehensive analysis of LoRA's viability as a practical LLM memory, investigating critical properties including its fundamental capacity, data efficiency, and extendibility to multi-LoRA scenarios.

Concurrently, (Schulman & Lab, 2025) shares our interest in analyzing the optimized settings, capacity, and efficiency of LoRA; however, while they focus on task LoRA learning curves compared to full fine-tuning, we distinguish our work by exploring LoRA specifically as a long-context memory.

**LLM Memory.** A central challenge in continual learning is enhancing the memory capacity of large language models (LLMs). In-context learning (ICL) (Brown et al., 2020) offers a temporary mechanism but is constrained by window length. To overcome this, system-level solutions introduce non-parametric external memory modules that can be dynamically written and retrieved, as seen in Xu et al. (2025) and Chhikara et al. (2025). Conversely, parametric approaches aim to internalize knowledge directly into model weights; for instance, WISE (Wang et al., 2024) duplicates full network layers to handle sequential editing. Advancing further into long-context capabilities, strategies such as Behrouz et al. (2024; 2025); Dao & Gu (2024) attempt to extend memory through architectural adaptation. Distinguishing our work, we leverage LoRA to instantiate a parameter-efficient plug-and-play memory module for long-context scenarios, specifically focusing on quantifying the static capacity of this modular memory.

**Retrieval-Augmented Generation.** Retrieval-Augmented Generation (RAG) (Lewis et al., 2020) addresses memory limitations by retrieving relevant external documents and conditioning generation on them. RAG has proven highly effective for many tasks, but its embedding-based retrieval mechanism has inherent representational limitations (Weller et al., 2025). Moreover, top-$k$ retrieval restricts the accessible context, which poses challenges for tasks requiring holistic integration of information spread across an entire document (Kuratov et al., 2024).

**Multi-LoRA and Parameter-Space Interpolation.** Another emerging line of work investigates the combination of multiple LoRA modules. Early explorations have shown that interpolating LoRA parameter spaces can compose multiple specialized capabilities (Huang et al., 2023; Feng et al., 2024; Dou et al., 2024; Prabhakar et al., 2025). Recent scalability efforts, such as LAG (Fleshman & Van Durme, 2025), focus on the routing challenge, proposing efficient retrieval mechanisms to select appropriate modules from libraries containing thousands of LoRAs. Concurrently, PRAG (Su et al., 2025) and DyPRAG (Tan et al., 2025) introduce 'Parametric RAG' frameworks, where document-specific LoRAs are pre-trained or dynamically generated, and then merged at inference using techniques like TIES-Merging. While these frameworks primarily focus on system-level architectures utilizing short Wikipedia contexts, our work complements them by offering a foundational analysis of LoRA as a long-context memory. We rigorously investigate intrinsic storage properties, including capacity limits, optimal data formatting, and merging scalability.

**Synthetic Data.** Recent work (Lampinen et al., 2025; Park et al., 2025; Lin et al., 2025; Yang et al., 2025b; Zhang et al., 2024) highlights the importance of data quality and structure for fine-tuning LLMs. Beyond directly training on raw data, studies show that transforming source material into diverse synthetic formats—such as QA pairs, summaries, or rewrites—can improve generalization and robustness. These findings suggest that synthetic augmentation serves as an effective strategy for building denser, more task-aligned knowledge representations.

# B  PHONEBOOK BENCHMARK

The PhoneBook benchmark was created to provide a controlled environment for evaluating a model's ability to memorize and recall novel, symbolic associations. This synthetic dataset is designed to be entirely disconnected from the knowledge embedded in the base model during pre-training, thereby isolating the model's capacity for new learning. Its two primary design goals are: (1) to test the pure memorization of arbitrary key-value pairs (fictional name-to-phone number mappings), and (2) to be programmatically scalable, allowing for precise control over the volume of knowledge used in our capacity and efficiency analyses (Section 3).

## DATA GENERATION AND COMPOSITION

The foundation of the benchmark is a source file, `synthetic_phonebook.csv`, containing a large set of unique, fictional name-phone number pairs. These pairs were generated programmatically to ensure that no real-world entities were included, thus preventing any potential knowledge leakage from the model's pre-training data. Fictional names were synthesized by combining common first and last names, and phone numbers were generated in a standard North American format (e.g., `(XXX) XXX-XXXX`).

From this source file, we generate training data by converting each name-number pair into a structured Question-Answering (QA) format. This approach frames the information as a natural language question and a direct answer, providing a clear and explicit learning signal for the model. An example of the QA format is shown below.

```
Question: What is the phone number of John Doe? Answer: 123-456-7890
```

## SCALABLE SLICING BY TOKEN COUNT

A key feature of the PhoneBook benchmark is its generation of multiple dataset "slices" with varying token counts. This allows us to precisely control the amount of information a LoRA module is trained on.

For each target size (e.g., 4K tokens), we iterate through the master source file in a fixed order. Each name-number pair is formatted into the QA format and tokenized using the `Qwen/Qwen3-8B` tokenizer. The formatted entry is added to the dataset slice, and its token count is added to a running total. This process continues until the total token count first exceeds the target size. This deterministic process ensures that smaller datasets are always perfect subsets of larger ones, providing a controlled and consistent methodology for studying the scaling properties of LoRA memory.

## EVALUATION PROTOCOL

Performance on the PhoneBook benchmark is measured using a strict **Exact Match (EM)** score. For evaluation, the model is presented with a question asking for the phone number of a specific name included in the training data. The model's generated output is then compared to the ground-truth phone number. A prediction is considered correct only if it is an exact, character-for-character match to the target answer. This strict criterion ensures that we are measuring precise recall, free from any partial credit or semantic ambiguity.

```
Question: "What is the phone number of Jane Smith?"

Ground Truth Answer: "987-654-3210"

# Correct Prediction (EM Score = 1)
Model Output: "987-654-3210"

# Incorrect Prediction (EM Score = 0)
Model Output: "The phone number for Jane Smith is (987) 654-3210."

# Incorrect Prediction (EM Score = 0)
Model Output: "(987) 654-3210"
```

## C    EXPERIMENTAL DETAILS OF SECTION 3

In this section, we provide the detailed experimental setup for the analysis presented in Section 3, which investigates the capacity and efficiency of LoRA modules.

**Base Model.** For all experiments, we used the `meta-llama/Llama-3.1-8B-Instruct` model available through the Hugging Face Transformers library as the base large language model.

Unless explicitly stated otherwise, this model served as the base for all other experiments presented in this paper.

**Training Hyperparameters.** We employed a consistent set of hyperparameters for training the LoRA modules to ensure a fair comparison across different configurations. The training was conducted for a total of 1500 steps with a batch size of 8. For the LoRA-specific configuration, we set the scaling factor $\alpha$ to be equal to the rank ($\alpha$=rank), which is a common practice that helps balance the influence of the low-rank adaptation.

**Parameter Sweep.** To thoroughly evaluate the effects of LoRA rank and data size on performance, we conducted a systematic sweep over these two dimensions. We varied the LoRA rank (r) across a range of values, doubling it at each step, with the specific ranks used in our experiments being 2, 4, 8, 16, 32, 64, 128. We also systematically increased the size of the training data from 1,000 tokens to 16,000 tokens, with an increment of 1,000 tokens for each experimental run. This allowed us to observe how performance scales with the amount of information stored in the LoRA module.

## D   DETAILS OF Q1. IS MEMORY CAPACITY SCALABLE WITH LORA RANK?

### MOTIVATION

The viability of employing Low-Rank Adaptation (LoRA) as a knowledge memory module for Large Language Models (LLMs) hinges on a fundamental question: is its capacity scalable? If the amount of information LoRA can store is severely limited or cannot be reliably controlled, its utility as a memory architecture is questionable. For researchers and practitioners, the LoRA rank is the most direct and accessible hyperparameter for modulating model capacity. However, a systematic analysis of the explicit relationship between LoRA rank and memory capacity has been notably absent in prior work. Understanding this relationship is crucial for addressing the practical question: *What rank is required to store a given amount of knowledge?* Answering this question will establish foundational guidelines for the principled design and efficient resource allocation of LoRA-based memory modules, moving beyond ad-hoc selection of the rank parameter.

### EXPERIMENTAL SETUP

We use CF dataset with 10K tokens length. For other experimental setups, we followed Appendix C.

### EXPERIMENTAL RESULTS

Across both the PhoneBook and CounterFact datasets, we observe a consistent and positive correlation between the LoRA rank and the model's performance. As illustrated in Figure 1 (left), performance on the CounterFact-10k dataset shows a distinct monotonic increase as the rank is scaled from 2 to 1024. This trend demonstrates that allocating more parameters to the LoRA module—by increasing its rank—directly enhances its capacity to reliably internalize a larger or more complex body of information.

However, we also observe a pattern of diminishing returns. The marginal performance gain for each incremental increase in rank tends to decrease, especially at higher rank values. This suggests that while capacity consistently grows with rank, the parameter efficiency for storing new knowledge diminishes. This trade-off between absolute capacity and efficiency is a critical consideration for the practical application of LoRA as a memory module.

## E   DETAILS OF Q2. IS THERE A FINITE CAPACITY LIMIT? HOW IS IT DETERMINED?

### MOTIVATION

While our initial findings confirm the scalability of LoRA's memory capacity with rank, this scalability cannot be infinite. For LoRA to be reliably integrated into practical systems as a memory component, it is imperative to move beyond observing scalability and instead identify its finite capacity limits. Understanding where these limits lie and how they are determined by the model's

... 

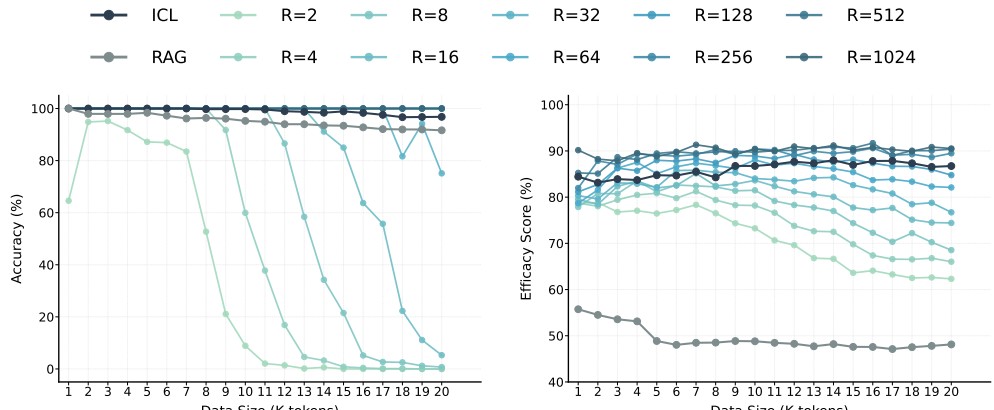

Figure 7: **Left:** Performance on PhoneBook for various ranks as data length increases. **Right:** Performance on CounterFact for various ranks as data size increases.

architecture is not merely a theoretical question. It has direct and critical implications for system design, informing how to provision resources and predict model behavior when faced with varying knowledge loads. This investigation seeks to answer: *At what point does a LoRA module of a given rank become saturated, and what governs this saturation point?*

EXPERIMENTAL SETUP

To empirically determine the capacity limits, we designed an experiment that jointly varies the LoRA rank and the volume of training data. We conducted a comprehensive grid search over these two primary variables on both the PhoneBook (PB) and CounterFact (CF) datasets.

- **LoRA Rank ($r$):** The rank was varied exponentially across 10 distinct values, from 2 to 1024 ($r \in \{2, 4, 8, \ldots, 1024\}$).

- **Data Volume:** For each rank, the number of training data tokens was varied linearly across 20 steps, from 1,000 to 20,000, in increments of 1,000 tokens.

All other hyperparameters and experimental conditions adhere to the configurations detailed in the Appendix C. This setup allows us to observe the performance of each rank configuration as it is exposed to an increasing amount of information, thereby revealing its saturation point.

EXPERIMENTAL RESULTS

Our experiments reveal that a LoRA module possesses a clear, finite memory capacity, and this limit is fundamentally determined by its rank. We observe a distinct **saturation phenomenon** across both datasets. For any given rank, performance remains high or increases as the data volume grows, but only up to a certain threshold. Beyond this point, performance degrades sharply, indicating that the module's capacity has been exceeded.

Crucially, this saturation point is a direct function of the rank. As depicted in Figure 7, higher-rank LoRA modules are capable of internalizing a significantly larger volume of knowledge before their performance collapses. This explicitly shows that a larger rank effectively raises the **capacity ceiling**.

These results yield a critical guideline for practitioners: the total volume of knowledge to be stored must be carefully matched with the allocated LoRA rank. Attempting to inject a large corpus of information into a low-rank LoRA module will inevitably lead to performance degradation due to this capacity overflow. Therefore, developers must provision a rank sufficient for the target knowledge volume to ensure stable and successful knowledge internalization.

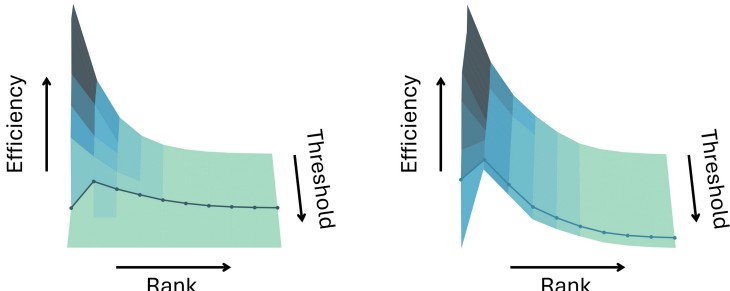

Figure 8: Surface generated by rank, threshold and efficiency. **(Left)** CounterFact. **Right** Phone-Book.

# F    DETAILS OF Q3. IS USING THE HIGHEST RANK ALWAYS THE MOST EFFICIENT CHOICE?

MOTIVATION

The findings from our previous sections confirm that LoRA's capacity scales with rank, which might suggest a straightforward strategy: use the highest rank available within resource constraints. However, this *more is better* approach requires rigorous engineering validation from a cost-benefit perspective. This section challenges this naive assumption by investigating whether the benefit of increased storage capacity is always proportional to the cost of a higher rank (i.e., more parameters, longer training times, and a larger memory footprint). If a regime of **diminishing returns** exists—where doubling the rank fails to double the effective capacity—then indiscriminately increasing the rank is a suboptimal strategy.

To formalize this, we move beyond raw performance and introduce a quantitative metric for efficiency: **Parameter Efficiency**, defined as the amount of information stored per parameter. This metric allows for a fair and objective comparison between LoRA modules of different ranks, enabling us to answer the question of how much knowledge can be stored per unit of cost. By analyzing this, we can identify an optimal design specification, or a *sweet spot*, for a single LoRA module, revealing the rank at which it is most efficient at compressing and internalizing new information.

EXPERIMENTAL SETUP

To evaluate parameter efficiency, we systematically vary the LoRA rank and measure its corresponding storage capability. We set the LoRA rank $r$ to span exponentially from 2 to 1024 ($r \in \{2, 4, 8, \ldots, 1024\}$). For each rank, we measure the **Parameter Efficiency** using the formulation from Equation 1 across various performance thresholds. This allows us to map out the efficiency landscape as a function of rank.

EXPERIMENTAL RESULTS

Our results provide an unequivocal answer: **No, the highest rank is not the most efficient choice.** The efficiency measurements, plotted in Figure 2, reveal that the parameter efficiency curve is non-monotonic. Instead of increasing with rank, the curve peaks at a specific point and subsequently declines. This indicates that beyond an optimal point, the growth in parameter count outpaces the effective gains in memory capacity, leading to decreased overall efficiency.

Specifically, across both the PhoneBook and CounterFact datasets, we observe peak storage efficiency in the low-rank regime. The most efficient configuration was consistently found at a small rank, such as $r = 4$. This finding strongly suggests that smaller, more compact LoRA modules can be significantly more parameter-efficient for knowledge storage than their larger counterparts.

# G  PAPERQA BENCHMARK

The PaperQA benchmark is designed to rigorously evaluate a model's ability to internalize and reason over novel, complex information. The construction of this benchmark is guided by three core principles: ensuring the novelty of the knowledge, generating a comprehensive set of evaluation questions, and establishing a sophisticated evaluation protocol.

**Source Material and Knowledge Novelty.** To ensure the novelty of the knowledge, we selected source materials from recent top-tier academic conferences. Specifically, we chose five oral or spolt-light papers from each of NeurIPS 2024, ICLR 2025, and ICML 2025. We extracted the introduction section from each paper to serve as the knowledge source. Crucially, we verified that the base model had no prior knowledge of these contents before the experiments. By using recent academic papers, we can assess the model's ability to update its factual knowledge and apply its understanding of complex information, allowing for a more precise evaluation of its learning capabilities.

**Comprehensive Question Generation.** To ensure a comprehensive evaluation, we generated a set of questions for each document that spans three distinct cognitive levels: (1) Key Information Recall, which tests the retrieval of specific facts; (2) Contextual Comprehension, which assesses the understanding of information in its surrounding context; and (3) Logical Structure Inference, which evaluates the ability to deduce the underlying logical flow and relationships within the text. For each of the 15 papers, we created 10 questions for each cognitive level, resulting in a total of 30 questions per paper. This hierarchical question design enables a multi-faceted assessment of knowledge comprehension at various depths.

**Dataset Construction and Evaluation Protocol.** Through this process, we constructed a dataset consisting of 450 question-answer pairs for the 15 new academic papers. To evaluate the model's responses, we employed an LLM-as-a-judge approach, utilizing GPT-4.1 to score the responses with a detailed rubric on a 0-10 scale. This method provides a sophisticated metric that measures the degree of knowledge internalization, moving beyond simple correctness to offer a nuanced assessment of the model's understanding.

## G.1  PAPERS IN PAPERQA

ICML 2025

1. Monte Carlo Tree Diffusion for System 2 Planning
   https://arxiv.org/abs/2502.07202

2. An Analysis for Reasoning Bias of Language Models with Small Initialization
   https://arxiv.org/abs/2502.04375

3. Training Dynamics of In-Context Learning in Linear Attention
   https://arxiv.org/abs/2501.16265

4. Inductive Moment Matching
   https://arxiv.org/abs/2503.07565

5. Statistical Test for Feature Selection Pipelines by Selective Inference
   https://arxiv.org/abs/2406.18902

NEURIPS 2024

6. Human Expertise in Algorithmic Prediction
   https://arxiv.org/abs/2402.00793

7. Enhancing Preference-based Linear Bandits via Human Response Time
   https://arxiv.org/abs/2409.05798

8. Rho-1: Not All Tokens Are What You Need
   https://arxiv.org/abs/2404.07965

9. Learning Action and Reasoning-Centric Image Editing from Videos and Simulations
   https://arxiv.org/abs/2407.03471

10. The Value of Reward Lookahead in Reinforcement Learning
    https://arxiv.org/abs/2403.11637

ICLR 2025

11. Artificial Kuramoto Oscillatory Neurons
    https://arxiv.org/abs/2410.13821

12. Exploring the Loss Landscape of Regularized Neural Networks via Convex Duality
    https://arxiv.org/abs/2411.07729

13. In Search of Forgotten Domain Generalization
    https://arxiv.org/abs/2410.08258

14. Programming Refusal with Conditional Activation Steering
    https://arxiv.org/abs/2409.05907

15. Efficient and Accurate Explanation Estimation with Distribution Compression
    https://arxiv.org/abs/2406.18334

## G.2 GENERATION PROMPTS

```
You are an expert academic assistant tasked with creating a high-quality question-
    answering dataset from a research paper's introduction. Your goal is to generate 30
    question-and-answer pairs based exclusively on the provided text.

Instructions and Rules:
Source Grounding: All questions and answers MUST be derived solely from the provided
    introduction text. Do not use any external knowledge or make assumptions beyond what
    is written.

Question Hierarchy: You must create questions across three distinct levels of
    understanding, as defined below.

Quantity: Generate exactly 30 pairs in total: 10 for Level 1, 10 for Level 2, and 10 for
    Level 3.

Output Format: The output must be a single, valid JSON array of objects. Do not include
    any explanatory text, comments, or markdown formatting before or after the JSON code
    block.

Question Level Definitions:
Level 1: Key Information Recall (10 Questions)

Objective: To test the recall of specific, explicitly stated facts, proper nouns,
    terminology, and figures from the text.

Question Type: "What is...?", "What are the names of...?", "Which X was mentioned...?"

Level 2: Contextual Comprehension (10 Questions)

Objective: To test the understanding of relationships between concepts, such as cause-and-
    effect, problem-solution, comparisons, and the function of a component.

Question Type: "Why does...?", "What is the effect of A on B?", "How does X work?", "What
    is the difference between A and B?"

Level 3: Logical Structure Inference (10 Questions)

Objective: To test the understanding of the overall logical flow of the text, including
    identifying the core problem, the research gap, the proposed solution, and the main
    contribution.

Question Type: "What is the core problem the authors aim to solve?", "What research gap
    does this paper intend to fill?", "What is the main advantage of the proposed method
    ?", "Summarize the key contribution of this work in relation to prior limitations."

Desired JSON Output Format:
The final output MUST be a JSON array containing 30 objects. Each object must have three
    keys: level (integer: 1, 2, or 3), question (string), and answer (string).

Example:

[
  {
    "level": 1,
    "question": "What is the full name of the algorithm the authors integrate their
        estimator into?",
```

```
    "answer": "The authors integrated their estimator into the Generalized Successive
        Elimination algorithm."
  },
  {
    "level": 2,
    "question": "According to the text, what is the inverse relationship between response
        time and preference strength?",
    "answer": "The text states that users who strongly prefer to skip a product tend to do
        so quickly, while longer response times can indicate weaker preferences."
  },
  {
    "level": 3,
    "question": "What is the core reason complex psychological models are impractical for
        real-time systems, and how does this paper's proposal address it?",
    "answer": "They are impractical because they rely on computationally intensive methods
        like hierarchical Bayesian inference and MLE. This paper addresses it by
        proposing a computationally efficient method that frames utility estimation as a
        linear regression problem."
  }
]

Introduction Text to Analyze:
[INSERT INTRODUCTION TEXT HERE]

Now, generate the 30 Q&A pairs in the specified JSON format based on the text provided
    above.
```

## G.3   JUDGING PROMPTS

```
You are an impartial AI assistant acting as an expert judge. Your task is to evaluate a
    candidate's answer to a question about a technical document. Compare the candidate's
    answer against the gold standard answer.

[EVALUATION CRITERIA]
1.  **Factual Alignment**: Does the candidate answer state the same facts as the gold
    answer? It must not contradict the gold answer.
2.  **Completeness**: Does the candidate answer include all the key information and
    nuances present in the gold answer?
3.  **Relevance**: Is the answer focused and on-topic? It must not contain irrelevant or
    hallucinatory information.

[SCORING RUBRIC (0-10 SCALE)]
- **10**: Perfect. The candidate answer is factually identical to the gold answer,
    complete, and contains no extraneous information.
- **7-9**: Mostly Correct. The answer is factually correct but might omit a minor detail
    or be slightly verbose. The core information is present and accurate.
- **4-6**: Partially Correct. The answer has the right general idea but contains a
    significant factual error, a major omission, or irrelevant information.
- **1-3**: Incorrect. The answer is on-topic but factually wrong.
- **0**: Completely Incorrect. The answer is nonsensical, irrelevant, or fails to address
    the question.

[TASK]
Evaluate the [CANDIDATE ANSWER] based on the criteria above and its alignment with the [
    GOLD ANSWER]. Provide your output in a single JSON object with two keys: "score" (an
    integer from 0-10) and "rationale" (a brief, one-sentence explanation for your score).

[QUESTION]
{question}

[GOLD ANSWER]
{gold_answer}

[CANDIDATE ANSWER]
{predicted_answer}
```

# H  DETAILS OF Q4. HOW DOES SYNTHETIC DATA ENHANCE SINGLE LoRA'S KNOWLEDGE MEMORIZATION?

## MOTIVATION

Our analysis has established that a LoRA module possesses a finite memory capacity. Training such a module on raw text alone presents a challenge, as the model may struggle to discern which information is critical, akin to learning from a textbook without highlighted key concepts or practice questions. This raises a fundamental question: how can we refine raw data into more effective learning signals to maximize the utility of LoRA's limited capacity? This section investigates strategies for transforming source documents into synthetic data for more efficient knowledge internalization.

We explore two primary hypotheses. First, we consider the impact of knowledge *density* and *format*. It is plausible that different data structures, such as question-answer (QA) pairs or summaries, may offer more compressed and potent learning signals than unstructured narrative text. Second, we examine the scalability of synthetic data generation. While prior studies have indicated the benefits of synthetic data, it is not well-established whether performance scales monotonically with the quantity of synthetic data derived from a single source. Answering these questions can provide practical guidelines for constructing an optimal data processing pipeline for LoRA-based knowledge memorization.

## EXPERIMENTAL SETUP

**Synthetic Data Generation.** We compare three synthetic data generation strategies against a raw text baseline: **(1) Question-Answering (QA)**, **(2) Summary**, and **(3) Rewrite**. To analyze the impact of data scaling, we experimented with varying quantities for each strategy: 10, 20, and 40 pairs for QA; 2, 4, and 8 variations for Summary; and 1, 2, and 4 variations for Rewrite. All synthetic data was generated using GPT-4.1.

**Hyperparameter Configuration.** We used a fixed set of LoRA hyperparameters for all training runs to ensure a fair comparison: a rank of 16, an alpha of 16, a learning rate of 5e-5, and a total of 1000 training steps. Unless otherwise specified, these same hyperparameters were also applied to the single LoRA experiments on the PaperQA benchmark.

## EXPERIMENTAL RESULTS

The results suggest that transforming raw text into structured, synthetic data is a highly effective strategy for enhancing knowledge memorization in LoRA.

As shown in Figure 3 (left), all three synthetic data formats—QA, Summary, and Rewrite—resulted in markedly better performance than the raw text baseline. This observation supports the hypothesis that data augmentation provides a clearer and more potent learning signal for the model.

Among the different formats, those involving information compression appeared to be the most effective. The observed order of performance, from most to least effective, was **QA**, followed by **Summary**, **Rewrite**, and finally the raw text baseline. The QA and Summary formats, which distill key information, outperformed the less structured Rewrite and raw text formats. The QA format's top performance may be partially attributed to its structural alignment with the evaluation task, which is also QA-based. This suggests that consistency between the training data format and the target task is an important factor.

Finally, while performance tended to improve with a larger quantity of synthetic data for all methods, the efficiency of this improvement varied. The performance gain per token was highest for the QA format, followed by Summary and then Rewrite. This finding implies that the method of data augmentation and the quality of the resulting data may be more critical than the sheer volume of data used.

## H.1 GENERATION PROMPTS FOR QA

```
You are an expert question-answer pair generator.
Based on the text provided, generate exactly {num_samples} question-answer pairs.
Your output must be ONLY a single JSON object with a key "items" that contains a list of
    objects. Do not include any text outside the JSON. Each object in "items" must have
    two keys: "question" and "answer".

[TEXT]
{text}
```

## H.2 GENERATION PROMPTS FOR SUMMARY

```
Based on the text provided, generate exactly {num_samples} distinct summaries of the
    content.
Your output must be ONLY a single JSON object with a key "items" that contains a list of
    objects. Do not include any text outside the JSON. Each object in "items" must have a
    single key: "summary".

[TEXT]
{text}
```

## H.3 GENERATION PROMPTS FOR REWRITE

```
Based on the text provided, generate exactly {num_samples} distinct rewrites of the
    passage.
While keeping entities and key details intact, use different vocabulary and sentence
    structures.
Ensure that the length of each rewrite is approximately the same as the original text (aim
    to maintain similar word count and overall length).
Your output must be ONLY a single JSON object with a key "items" that contains a list of
    objects. Do not include any text outside the JSON. Each object in "items" must have a
    single key: "rewrite".

[TEXT]
{document_text}
```

# I DETAILS OF Q5. WHAT ARE THE EFFECTS OF COMBINING SYNTHETIC DATA FORMATS?

### MOTIVATION

The previous section identified the Question-Answering (QA) format as a particularly effective single method for data augmentation. This leads to a natural follow-up question: can combining different formats yield synergistic effects that surpass the performance of the best single format? Investigating this is essential for exploring new possibilities to move beyond single-method optimization and potentially achieve higher performance ceilings.

Furthermore, this inquiry offers a deeper understanding of LoRA's learning mechanisms. Different data formats may encourage the learning of distinct cognitive skills; for example, QA might enhance factual retrieval, summaries could foster high-level conceptual understanding, and rewrites might improve stylistic flexibility. By combining these formats, we can observe whether LoRA can integrate these varied learning signals in a complementary fashion. While the principle of data diversity is often considered beneficial, its specific scaling effects in the context of synthetic data for LoRA-based memorization have not been systematically explored. This study aims to address this by examining how performance responds to an increasing variety of data formats.

EXPERIMENTAL SETUP

This experiment is a direct continuation of the analysis in the previous section, utilizing the same synthetic datasets. We constructed new, mixed-format training datasets by concatenating the datasets generated previously. For instance, we used the dataset composed of 40 QA pairs from the prior experiment, as well as others like the one containing 8 summaries. A combined dataset was then created by merging the text files of these individual datasets, such as the 8-summary set and the 40-QA-pair set.

To isolate the effect of data composition, all other experimental conditions were held constant and remained identical to the previous section. This includes the base model (Llama3.1-8B-Instruct), LoRA hyperparameters (rank 16, alpha 16), and all training-related hyperparameters.

EXPERIMENTAL RESULTS

To investigate the synergistic effects of combining different synthetic data formats, we constructed mixed training datasets using the data generated in the experiment detailed in Section 4. Specifically, we utilized the largest-quantity datasets from each category: the 40 pairs from the Question-Answering (QA) set, the 8 variations from the Summary set, and the 4 variations from the Rewrite set. These individual datasets were combined by simple concatenation of their respective text files to create new, mixed-format training sets.

## J DETAILS OF Q6. HOW DOES THE SCALE OF BASE MODELS AFFECT LORA KNOWLEDGE INTERNALIZATION?

MOTIVATION

LoRA does not operate in isolation; it functions as an adapter that attaches to a pre-trained base model. Consequently, its final performance is likely influenced not only by its own parameters but also by the intrinsic capabilities of the underlying model. This dependency raises a critical question regarding the interplay between the adapter and the base model: does a larger, more capable base model provide a more fertile foundation for LoRA to effectively integrate and leverage new knowledge?

Understanding this relationship is crucial for assessing the practical deployment and cost-effectiveness of LoRA-based memory systems. The degree to which LoRA's effectiveness depends on the base model's scale helps to clarify the performance trade-offs involved in system design. If substantial performance gains are only achievable with large-scale models, the applicability of this method might be better suited for high-resource environments. Conversely, if LoRA can significantly augment the capabilities of smaller models, it presents a viable strategy for achieving competent performance within more constrained resource budgets.

EXPERIMENTAL SETUP

To isolate the effect of model scale on LoRA's knowledge internalization, we selected various models from the Qwen3 family, available through the Hugging Face Transformers library. The specific models used were `Qwen/Qwen3-0.6B`, `Qwen/Qwen3-1.7B`, `Qwen/Qwen3-4B`, `Qwen/Qwen3-8B`, and `Qwen/Qwen3-14B`. This approach allowed us to directly attribute performance differences to the inherent capacity of the base model. To ensure that the base model's scale was the sole variable, all other experimental conditions were held constant. Each model was fine-tuned using the raw text from the PaperQA benchmark as training data. The LoRA hyperparameters were fixed across all experiments, with a rank of 16.

EXPERIMENTAL RESULTS

Our results indicate a general trend where performance improves as the size of the base model increases. As depicted in Figure 4 (left), the PaperQA score consistently rises with the parameter count, from the 0.6B to the 14B model. This suggests that larger models may indeed provide a more advantageous foundation for internalizing and applying new knowledge.

However, this performance enhancement is not directly proportional to the model's scale and exhibits a notable non-linear pattern. We observed significant performance gains when scaling from 0.6B to 1.7B and again from 8B to 14B. In contrast, the performance improvement was relatively marginal across the intermediate-sized models, specifically from **1.7B to 8B**.

This non-linear relationship may offer insights into LoRA's operational mechanics. Unlike methods such as in-context learning, which heavily leverage the base model's intrinsic reasoning abilities, LoRA appears to focus more on directly storing new knowledge within its own added parameters. This could imply a relatively lower dependency on the base model's scale compared to other methods. Furthermore, it is noteworthy that even a relatively small model, such as the 1.7B variant, achieved a substantial level of knowledge processing capability when augmented with LoRA.

These findings have important implications for practical applications. The non-linear scaling suggests that a cost-benefit analysis is crucial when selecting a base model. For instance, in an operational range corresponding to our observed plateau region (1.7B to 8B), upgrading to a larger base model may yield only minimal returns. In such cases, focusing on improving data quality or tuning LoRA's rank could be a more cost-effective optimization strategy. While using the largest available model can generally be expected to yield higher performance, our results suggest that combining a well-optimized LoRA with a mid-sized model could be a more rational and efficient alternative for achieving specific performance targets.

# K DETAILS OF Q7. DOES SYNTHETIC DATA GENERATOR QUALITY IMPACT LoRA PERFORMANCE?

## MOTIVATION

The quality of training data is a foundational factor that often determines final model performance in machine learning. This section investigates the hypothesis that for synthetic data, the quality of the data-generating model directly influences the quality of the resulting training data and, consequently, the performance of the trained LoRA module.

This inquiry is also motivated by a practical need for a clear cost-benefit analysis. High-performance models such as GPT-4 incur significant costs for data generation, whereas open-source models like LLaMA offer a more cost-effective alternative. By quantifying the performance difference, this study aims to provide developers with practical guidance: does the investment in a superior, high-cost generator yield a proportional performance benefit, or can more accessible models produce data of sufficient quality?

## EXPERIMENTAL SETUP

To analyze the impact of the synthetic data generation model on LoRA's performance, we compared two distinct models: the high-performance, API-based GPT-4.1 and the accessible, local Llama-3.1-8B. For a fair and direct comparison, we standardized the amount of synthetic data to 20 QA pairs per document for both generation models. This adjustment was necessary because preliminary experiments showed that the Llama-3.1-8B model had difficulty reliably generating the larger set of 40 QA pairs used in other experiments. The LoRA hyperparameters were fixed with a rank of 16.

## EXPERIMENTAL RESULTS

The results indicate that the quality of the data-generating model appears to have a substantial impact on the final performance of the LoRA module. As shown in Figure 6, there is a clear performance disparity between the LoRAs trained on data from the two different generators. This gap seems to directly reflect the known capability difference between the generator models themselves. This performance difference can plausibly be attributed to the quality of the generated data; it is likely that the more advanced model produced training examples that were more accurate, logically coherent, and comprehensive.

This finding has a significant implication for the use of LoRA as a knowledge module. The performance of a knowledge-infused LoRA is not only dependent on its own architecture or training

process but is also deeply tied to the quality of its knowledge source. The process can be conceptualized as a form of knowledge *distillation*, where the knowledge contained within the large generator model is transferred to the more compact LoRA module. Consequently, the depth and breadth of knowledge that the generator model can comprehend and articulate may act as a practical upper bound on the knowledge that the LoRA module can effectively learn and represent.

## L   DETAILS OF Q8. HOW CAN WE UTILIZE MULTIPLE SMALL LoRAS TO OUTPERFORM A SINGLE LARGE LoRA?

### MOTIVATION

Our preceding analyses have characterized the fundamental properties of a single LoRA module. We have established that while its memory capacity is scalable with rank, it has discernible limits. Furthermore, we found that its parameter efficiency is often optimal in the low-to-mid rank regime. These findings motivate an alternative approach for managing large-scale knowledge, one that circumvents the limitations of a single, monolithic module. Specifically, the finite capacity of a single LoRA and the high parameter efficiency of low-rank modules suggest the potential advantages of a modular architecture: distributing knowledge across multiple small, efficient modules.

Therefore, this section conducts a proof-of-concept (PoC) to validate the core potential of this modular approach. To isolate the storage and retrieval benefits of the architecture itself, we perform this test within the PhoneBook (PB) dataset environment, where data length can be arbitrarily extended to simulate long-context challenges. Crucially, our initial analysis is performed under the assumption of an idealized setting with perfect routing. This allows us to evaluate the capacity of the modular system itself, free from any potential performance degradation that could be introduced by a separate routing mechanism.

### EXPERIMENTAL SETUP

This experiment was conducted on a 64K token PhoneBook dataset to compare three configurations under an identical parameter budget. The In-Context Learning (ICL) baseline was provided with the full 64K context at inference time. The Single LoRA configuration used a single module with a rank of 32, trained on the entire 64K dataset. For the Multi-LoRA configuration, the dataset was partitioned into eight 8K chunks, and a separate rank-4 module was trained on each chunk.

### EXPERIMENTAL RESULTS

The results of this prototype experiment (Figure 5, first) highlight the potential of the modular approach for handling large volumes of information. On this long-context task, the ICL method exhibited a significant performance drop, while the Single Large LoRA approach struggled to internalize the extensive knowledge base effectively.

In contrast, the Multi-LoRA system, operating under the perfect routing assumption, successfully learned the information within each respective chunk and demonstrated strong performance. While this outcome might be anticipated given the experimental design, its value lies not in its novelty but in its function as a clear proof of concept. This experiment serves to illustrate the conceptual advantages of a multi-LoRA architecture, where knowledge is partitioned and managed by specialized, efficient modules. Furthermore, it provides a foundational basis for discussing the critical components, such as the routing mechanism, that must be addressed to translate this concept into a practical, real-world system.

## M   DETAILS OF Q9. HOW MUCH PERFORMANCE LOSS DOES A MORE PRACTICAL ROUTING INCUR?

### MOTIVATION

The preceding proof-of-concept experiment demonstrated the potential of a multi-LoRA system under the idealized condition of perfect routing. However, real-world systems must make decisions

based on imperfect information. This section aims to quantify the performance gap between this ideal scenario and a more practical implementation using embedding-based retrieval. Measuring this gap is essential for understanding the performance cost imposed by real-world constraints.

Furthermore, this analysis helps to identify critical performance bottlenecks in a modular system. The final performance of a multi-LoRA system depends on multiple components, and this experiment investigates the extent to which the routing stage can limit the system's overall efficacy. If the performance loss from standard retrieval-based routing proves to be substantial, it would highlight the need for alternative strategies—such as merging multiple LoRAs—to mitigate the inherent uncertainties of the routing process.

### EXPERIMENTAL SETUP

To evaluate the impact of the routing mechanism, we compared the following three approaches.

**Perfect Routing (Oracle).** This serves as the theoretical upper bound for system performance. For each query, an oracle, with prior knowledge of which document chunk (e.g., a specific paper in the PaperQA dataset) contains the answer, selects the correct corresponding LoRA module.

**RAG-based Routing.** For the practical text-embedding-based routing scenario, we employed the `BGE-large-en-v1.5` (Chen et al., 2024) model to select LoRA modules based on embedding similarity between the query and source documents; this model is used for all subsequent embedding-based operations unless otherwise specified.

**Single-LoRA (Baseline).** For comparison, we also include the performance of a single, monolithic LoRA module trained on the entire dataset.

Apart from the routing method, all other experimental conditions, including the base model (Llama3.1-8B-Instruct), LoRA hyperparameters (rank 16, training hyperparameters) and training data, were kept consistent with previous PaperQA experiments (Appendix H) to ensure a controlled comparison.

### EXPERIMENTAL RESULTS

The experimental results show that the choice of routing mechanism is a critical factor in the performance of a multi-LoRA system. As expected, the **Perfect Routing** approach yielded the highest performance, which reaffirms that the modular approach can be highly effective if the correct specialized module is accurately identified.

In contrast, the **RAG-based Routing** method showed a noticeable performance degradation compared to the perfect routing oracle. This performance gap suggests that the retrieval-based routing mechanism is a primary limiting factor for the system's overall potential.

Perhaps the most notable finding is that the RAG-based Routing approach performed at a lower level than the **Single-LoRA** baseline. This outcome is particularly insightful, as it indicates that selecting an incorrect, highly specialized LoRA module can be more detrimental to performance than using a single, non-specialized module that contains the entirety of the knowledge base. This underscores the critical importance of routing accuracy in a modular architecture.

## N    DETAILS OF Q10. CAN MERGING MULTIPLE LoRAs MITIGATE ROUTING ERROR?

### MOTIVATION

The previous section established that retrieval-based, top-1 routing can be a significant performance bottleneck, highlighting the inherent risks of relying on a single module selection. This finding motivates the exploration of an alternative strategy to distribute this risk: retrieving the top-k relevant LoRA modules and merging their knowledge. This section aims to validate the feasibility of such a merging strategy as a direct approach to mitigating routing uncertainty.

Furthermore, LoRA merging is not a monolithic concept; it encompasses a design space ranging from simple arithmetic combinations to more sophisticated algorithms intended to mitigate interference between parameters. We investigate how different merging algorithms perform from a knowledge preservation perspective and analyze the reasons for these differences. This inquiry also touches upon a fundamental property of LoRAs: their *composability*. By arithmetically combining independently trained modules, we can assess whether the knowledge encoded in each can be preserved without destructive interference, thereby testing their viability as composable building blocks.

EXPERIMENTAL SETUP

We provides a detailed description of the three LoRA merging methods here: (1) Linear Averaging, (2) Matrix Concatenation (Cat), and (3) TIES-Merging.

**Linear Averaging.** Linear Averaging is the most intuitive and straightforward method for merging multiple LoRA modules. It operates by taking a weighted average of the parameters from each module. A new, single LoRA module is created by multiplying the parameter matrices ($W_i$) of each constituent LoRA module by a corresponding scalar weight ($\lambda_i$) and summing the results.

For merging three LoRA modules, the final merged weight, $W_{\text{merged}}$, is calculated as:

$$W_{\text{merged}} = \lambda_1 W_1 + \lambda_2 W_2 + \lambda_3 W_3$$

where $\sum_i \lambda_i = 1$. In our experiments, we employed a uniform averaging scheme for the top-3 retrieved LoRAs, assigning equal weights, i.e., $\lambda_1 = \lambda_2 = \lambda_3 = 1/3$. While this method is simple to implement, its primary limitation is its inability to account for potential conflicts or redundancies among the parameters of the different LoRA modules.

**Matrix Concatenation (Cat).** Matrix Concatenation (Cat) constructs a single, more expressive LoRA module by concatenating the respective low-rank matrices (A and B) from multiple source modules. This process effectively increases the rank of the final merged adapter. For instance, given three LoRA modules, each with rank $r$, their decomposed matrices can be represented as $A_i \in \mathbb{R}^{d \times r}$ and $B_i \in \mathbb{R}^{r \times k}$.

The Cat merge operation combines these matrices along a specified dimension:

$$A_{\text{merged}} = \text{concat}([A_1, A_2, A_3], \dim = 1)$$
$$B_{\text{merged}} = \text{concat}([B_1, B_2, B_3], \dim = 0)$$

The resulting merged LoRA module has a rank of $3r$, endowing it with a higher capacity for representation compared to any individual module. A key advantage of this approach is its ability to integrate and preserve the specialized knowledge learned by each LoRA.

**TIES.** TIES (TrIm, Elect sign, and Merge) (Yadav et al., 2023) is an advanced methodology designed to mitigate the interference that often occurs when merging parameters from multiple fine-tuned models. It specifically addresses two primary sources of interference: parameter redundancy and sign conflicts. The TIES-Merging procedure consists of the following three steps:

1. **Trim:** This step zeroes out parameters that underwent minimal change during the fine-tuning of each LoRA module. By retaining only the parameters with the largest change in magnitude (e.g., the top-k percentile), this process filters out redundant or less impactful parameters.

2. **Elect Sign:** This step resolves directional conflicts in parameter updates across different LoRA modules. For a given parameter, some LoRAs may have induced a positive update while others induced a negative one. TIES elects a single, dominant sign based on a majority vote, where the sign corresponding to the greatest total magnitude of updates is chosen as the consensus direction.

3. **Merge:** Finally, only the parameter values that align with the elected sign are averaged. Parameters with conflicting signs are discarded from the merge for that specific weight, thus minimizing negative interference.

Through this structured process, TIES effectively preserves the most salient changes from each LoRA module while resolving conflicts between them, leading to a more robustly performing merged model. All other experimental conditions, including the base model (Llama3.1-8B-Instruct), LoRA hyperparameters (rank 16), training hyperparameters and training data, were kept consistent with previous PaperQA experiments (Appendix H) to ensure a controlled comparison.

EXPERIMENTAL RESULTS

Our results indicate that the choice of merging algorithm critically affects the performance of the multi-LoRA system. According to Table X, the **TIES** merging method achieved the highest performance among all techniques tested. Its score surpassed that of the top-1 RAG-based routing approach and was nearly on par with the Single-LoRA baseline. This suggests that a sufficiently sophisticated merging algorithm can effectively compensate for the performance loss associated with routing imperfections.

In contrast, the **Linear** merge performed poorly, scoring lower than not only TIES but also the top-1 RAG-based routing baseline. This outcome suggests that naively combining the weights of multiple LoRAs can lead to a phenomenon of *parameter interference*, where the knowledge stored in individual modules is corrupted or destroyed.

The **CAT** method resulted in a near-total failure of the system. This is likely attributable to an architectural incompatibility; arbitrarily concatenating LoRA matrices appears to disrupt the model's operational integrity, preventing coherent computation.

## O  DETAILS OF Q11. HOW DOES THE NUMBER OF MERGED LORAS AFFECT PERFORMANCE?

MOTIVATION

In a system that merges the top-N retrieved LoRAs, the choice of N is a critical hyperparameter that mediates a trade-off between the *recall* of the routing phase and the *precision* of the merging phase. Understanding how system behavior changes with N is essential for designing and tuning practical multi-LoRA architectures.

This system architecture inherently faces two competing risks: (1) **routing failure**, where a small N may fail to retrieve the correct knowledge module, and (2) **merging failure**, where a large N may lead to knowledge corruption through parameter interference. Our previous experiments provided evidence for both: the superior performance of a top-3 TIES merge over a top-1 selection suggested that merging can mitigate routing failures, while the poor performance of linear merging hinted at the dangers of parameter interference. This section seeks to empirically map this trade-off by systematically varying N, providing a deeper analysis of how sensitively parameter interference affects the system as the number of merged modules grows.

EXPERIMENTAL SETUP

The experiment was designed to observe performance changes as we increased the number of merged LoRA modules, N, from 1 to 5. To isolate the effect of the merging process from routing errors, our evaluation for a given N involved merging the specific N LoRA modules that were trained on the N documents from which the test queries were sampled. This setup ensures that all necessary knowledge is present within the merged set, allowing us to measure only the degradation caused by the merging process itself.

Based on the findings from the previous section, we used the **TIES** algorithm for all merging operations. All other experimental conditions, apart from the value of N, were kept identical to the previous experiment on merging algorithms.

EXPERIMENTAL RESULTS

The results show a consistent and sharp decline in overall system performance as the number of merged LoRAs (N) increases. As illustrated in Figure X, the performance curve peaks at N=1 (which is equivalent to perfect top-1 routing) and then degrades steadily as N grows.

This outcome provides strong empirical evidence for the phenomena of *knowledge dilution* and *parameter interference*. The analysis suggests that as more LoRA modules—even relevant ones—are merged, the cumulative conflicts between their parameter values rapidly degrade system performance.

These findings reveal a distinct trade-off in multi-LoRA systems that is governed by the number of merged modules. A small N risks routing failure, whereas a large N is dominated by performance loss from parameter interference. This implies that a successful multi-LoRA system requires a more sophisticated strategy than simply merging a fixed number of top-retrieved modules. Future work should explore methods for dynamically adjusting N based on retrieval confidence or developing merging algorithms that are more robust to interference, thereby striking a more effective balance between routing and merging.

## P    NARRATIVEQA BENCHMARK

The NarrativeQA dataset is a challenging question-answering benchmark characterized by its exceptionally long source documents, which average approximately 60,000 tokens each. The official dataset is partitioned into 1,102 training documents, 115 development documents, and 355 test documents.

A distinctive and challenging feature of this dataset is its data generation and evaluation protocol. The question-answer pairs for each document are created based on human-written summaries of the entire text. However, the evaluation is performed exclusively against the original, full-length source document. This design makes the dataset particularly difficult for models with limited context windows, as the ground-truth answers may require the synthesis of information scattered across the entire narrative, far exceeding a typical context length.

For our case study, we focused on documents with token lengths ranging from 10,000 to 20,000 to maintain a focused experimental scope. We randomly sampled a subset of 40 documents and their corresponding question-answer pairs from the official training and validation sets.

To adapt these documents for our multi-LoRA system, each document was segmented into chunks of 2,048 (2K) tokens. We employed an overlap of 200 tokens between consecutive chunks to help preserve contextual continuity across chunk boundaries. Each of these chunks was then used to train an individual LoRA module.

## Q    DETAILS OF Q12. HOW DOES LoRA MEMORY PERFORM ON LONG-CONTEXT MULTI-HOP QA TASK?

### MOTIVATION

While our preceding experiments utilized datasets with explicitly partitioned corpora (e.g., Phone-Book, PaperQA), real-world scenarios often involve long-form documents without clear boundaries. To investigate this, we use the NarrativeQA (NQA) dataset (Kočiskỳ et al., 2018), which consists of long narratives where segmenting the text for individual LoRAs creates strong contextual dependencies between chunks. The NQA questions often require synthesizing information across these chunk boundaries, demanding a holistic understanding of the text. This inter-chunk dependency presents a critical challenge for modular approaches, as a multi-LoRA system may fail to capture the overarching narrative. This section analyzes this potential failure mode and its impact on performance.

### EXPERIMENTAL SETUP

For experiments on NQA, we used a subset of 40 documents and two base models: Llama-3.1-8B and Llama-3.2-1B. In our multi-LoRA setup, each document is segmented into 2K-token chunks, each used to train a distinct LoRA module. For all experiments, we maintained a fixed batch size of 32 and a learning rate of $5 \times 10^{-4}$. For the multi-LoRA modules, we employed a rank of $r = 4$ and a scaling factor $\alpha = 8$, training for 150 steps. In contrast, the single LoRA baseline was configured with $r = 16$ and $\alpha = 32$, and trained for 250 steps.. During inference, we evaluated two distinct retrieval strategies: composing the single most relevant LoRA module (Top-1) and merging the three most relevant modules (Top-3). We used TIES-merging method.

Although a Multi-LoRA system is conceptually well-suited for such long documents, the observed performance deficit is likely a consequence of the compounding error sources discussed in Sections Q9 and Q10. The dual challenges of routing inaccuracy and parameter interference are intensified by the nature of the NarrativeQA dataset. Its reliance on multi-hop questions, which demand the synthesis of information across multiple text segments, inherently strains the capabilities of our chunk-based modular approach.

# R    DETAILS OF Q13. CAN LoRA PERFORM BETTER WHEN USED WITH EXTERNAL CONTEXT?

## MOTIVATION

LoRA-based models store knowledge in a compressed format, which means they do not have access to the original, high-fidelity training data at inference time. This inherent characteristic can be a limitation. The challenge is further compounded in Multi-LoRA systems, This problem is exacerbated in multi-LoRA systems, where training across multiple modules can lead to a loss of contextual continuity and fragmented internal memory.

This raises a critical question: *can supplying explicit external context at inference time compensate for these limitations?* In this section, we investigate the potential of augmenting both Single and Multi-LoRA systems with external information. We hypothesize that such an approach will enhance overall performance by providing the models with precise, relevant information that may be absent or diluted in their internal representations. We further posit that this benefit will be particularly pronounced for Multi-LoRA configurations, as external context can help bridge the narrative gaps introduced by the module merging process.

## EXPERIMENTAL SETUP

To evaluate the interaction between LoRA and external context, we constructed several hybrid systems and benchmarked them against standalone baselines.

**Hybrid Systems.** : We explored two primary integration strategies. First, we paired a Single LoRA module with both In-Context Learning (ICL) and Retrieval-Augmented Generation (RAG). Second, given its architectural alignment with retrieval mechanisms, we combined our chunk-based Multi-LoRA system with RAG. For the Multi-LoRA hybrid, we evaluated two distinct merging strategies: one that merges the single most relevant LoRA module retrieved (Top-1) and another that merges the top three most relevant modules (Top-3).

**Baselines.** : The performance of these hybrid systems was compared against standalone ICL, standalone RAG, and the closed-book Single and Multi-LoRA configurations from our previous experiments.

## EXPERIMENTAL RESULTS

As presented in Table 1, the integration of LoRA with external context improves performance across all configurations. For both model scales, the hybrid systems outperform their standalone LoRA, ICL, and RAG counterparts.The performance gain is most pronounced for the Multi-LoRA system. This suggests that the provision of external information effectively compensates for the precision loss that can occur when merging multiple specialized LoRA modules.

Interestingly, we observe that ICL yields a greater performance uplift than RAG when combined with LoRA. A possible explanation is that the continuous, global context supplied by ICL is uniquely effective at restoring the narrative cohesion that may be lost among fragmented LoRA modules. The snippet-based retrieval approach of RAG, while beneficial, appears less capable of fully overcoming this particular challenge.

# S   DETAILS OF Q14. CAN MERGING LoRAs IMPROVE CONTEXTUAL CONTINUITY?

## MOTIVATION

In the previous section, we demonstrated that external context can help mitigate the loss of contextual continuity in Multi-LoRA systems. An alternative approach for complex reasoning involves the direct synthesis of knowledge from multiple LoRA modules. PRAG (Su et al., 2025) has shown that merging LoRAs can be effective for tasks requiring the integration of knowledge from several distinct sources. However, PRAG focused on scenarios where the knowledge is sourced from multiple, explicitly distinct, and relatively short documents. The question of whether such a merging strategy is beneficial for tasks demanding deep, continuous comprehension of a single long document has not been investigated. This motivates our current investigation: we re-evaluate the efficacy of the multi-LoRA merging strategy on the NarrativeQA dataset, which requires holistic document comprehension. This allows for a assessment of whether synthesizing knowledge from multiple modules offers a tangible advantage over relying on a single, best-matched module for multi-hop QA tasks.

## EXPERIMENTAL SETUP

The overall experimental setup is identical to Q12 and Q13.

## EXPERIMENTAL RESULTS

The results, presented in Table 1, reveal a clear and consistent performance advantage for the Top-3 merging strategy over the Top-1 approach. In the closed-book setting, the larger model in particular shows an improvement with the Top-3 configuration compared to the Top-1.

This performance gap becomes more pronounced in the open-book setting, where models are provided with external context. The Multi-LoRA Top-3 configurations, when paired with either ICL or RAG, consistently and significantly outperform their Top-1 counterparts. Notably, the combination of Top-3 merging with ICL achieves the highest performance across all tested methods.

This finding suggests that for tasks requiring multi-hop reasoning, accessing a broader set of knowledge sources by merging multiple modules is highly advantageous. The model appears to leverage the combined knowledge from several specialized LoRAs to construct more comprehensive and accurate answers, a benefit that is amplified by the presence of external guiding context.

# T   DETAILS OF Q15. HOW DOES LoRA BENEFIT IN TIME?

## MOTIVATION

While the main paper establishes the performance of LoRA-based memory systems in terms of accuracy, a comprehensive evaluation must also consider their practical viability from a computational standpoint. Context-based methods like ICL and RAG are known to incur significant latency due to the need to process long context windows for every query. In contrast, LoRA-based methods internalize knowledge into parameters, theoretically enabling much faster inference with short contexts, but introducing their own overheads such as module loading and merging.

The purpose of this experiment is to provide a granular, quantitative analysis of these computational trade-offs. By measuring the end-to-end latency and breaking down the time spent on each component—from model loading to final token generation—we can create a clear picture of the real-world costs and benefits associated with each approach. This section details the complete methodology for this analysis, beginning with the hardware and software environment, followed by the rigorous timing measurement protocol, and concluding with a precise definition of every component measured for each experimental condition.

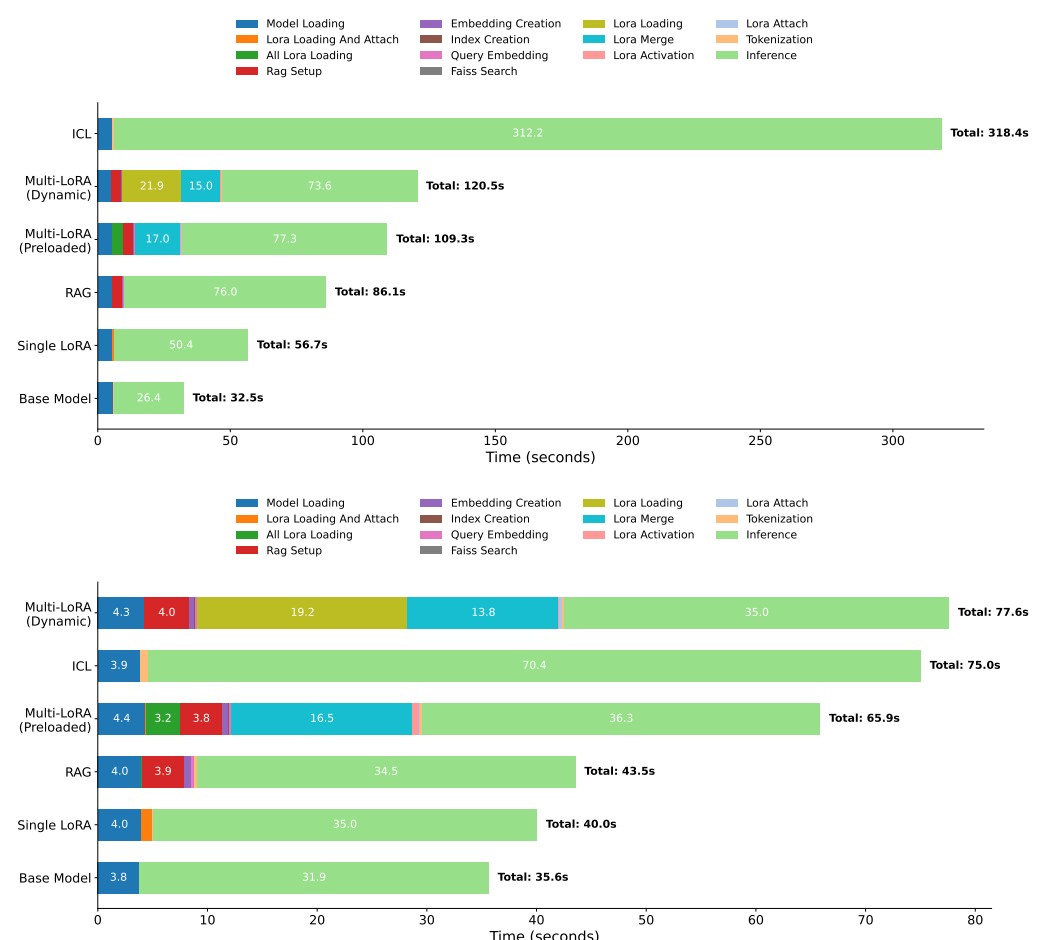

Figure 9: Comparison of method execution times. (**Top**) Without Flash Attention. (**Bottom**) With Flash Attention.

## EXPERIMENTAL SETUP

**Hardware and Software Specifications.** All experiments were conducted on RTX 6000 Pro. The core software stack included PyTorch for model operations, the Hugging Face `transformers` library for loading the base model, the `peft` library for handling LoRA modules, `sentence-transformers` for text embeddings, and `faiss` for efficient similarity search in retrieval tasks. The base model for all experiments was `meta-llama/Llama-3.1-8B-Instruct`, and the retrieval model was `BAAI/bge-large-en-v1.5`.

**Timing Measurement Protocol.** To ensure accurate and reliable timing, especially for GPU operations, we implemented a standardized measurement protocol. All timed operations were wrapped in a utility function that uses `time.perf_counter()` for high-precision timing. Crucially, before starting and after ending the timer, we explicitly called `torch.cuda.synchronize()`. This call blocks the CPU execution until all previously queued GPU kernels have completed, which is essential for accurately measuring the wall-clock time of asynchronous GPU computations and avoiding misleading results.

## METHODOLOGY AND MEASURED COMPONENTS

We systematically measured the time taken by each distinct stage of the inference process for six different methods. The total time for each method is the sum of its one-time setup costs and the cumulative time of all per-question operations over the 30 questions in the evaluation set.

BASE MODEL (CLOSED-BOOK)

This configuration measures the baseline performance of the LLM without any external context or adapters.

- **model loading**: One-time cost to load the weights of the `Llama-3.1-8B-Instruct` model into GPU memory.
- **tokenization**: Per-question time to convert the short prompt (question only) into input tokens.
- **inference**: Per-question time for the model to generate the answer tokens.

IN-CONTEXT LEARNING (ICL)

This method provides the model with the entire source document as context for every query.

- **model loading**: Same as the base model.
- **tokenization**: Per-question time to tokenize the prompt, which includes the **full document text** and the question. This is computationally more expensive than in the closed-book setting.
- **inference**: Per-question generation time. This is the most significant component due to the quadratic complexity of self-attention over the long context window.

RETRIEVAL-AUGMENTED GENERATION (RAG)

This method retrieves relevant text snippets to use as context.

- **model loading**: One-time cost to load the LLM.
- **rag setup**: A one-time setup cost that includes:
  - Loading the `bge-large-en-v1.5` embedding model.
  - **embedding creation**: Generating embeddings for all text chunks of the document.
  - **index creation**: Building the FAISS index from the chunk embeddings.
- **query embedding**: Per-question time to generate an embedding for the input question.
- **faiss search**: Per-question time to perform a similarity search against the FAISS index to retrieve the top-3 relevant chunks.
- **tokenization**: Per-question time to tokenize the prompt containing the retrieved chunks and the question.
- **inference**: Per-question generation time using the retrieved context.

SINGLE LoRA (CLOSED-BOOK)

This method uses a single LoRA module trained on the entire document.

- **model loading**: One-time cost to load the base LLM.
- **lora loading and attach**: A one-time setup cost to load the single LoRA adapter from disk and attach it to the base model using `PeftModel.from_pretrained`.
- **tokenization**: Per-question time to tokenize the short prompt (question only).
- **inference**: Per-question generation time.

MULTI-LoRA (PRELOADED)

This method preloads all LoRA modules for a document into GPU memory at the start and merges the relevant ones for each query. This strategy is designed to minimize I/O latency during inference.

- **model loading**: One-time cost to load the base LLM.

- **all_lora_loading**: A one-time setup cost to load **all** LoRA modules corresponding to the document's chunks into GPU memory.

- **rag_setup**: Same as the RAG method, used for retrieving relevant LoRA modules.

- **query_embedding** and **faiss_search**: Per-question retrieval time to identify the top-3 relevant LoRA modules.

- **lora_merge**: Per-question time to combine the weights of the top-3 retrieved LoRA adapters into a new, temporary adapter using the TIES-merging algorithm (`model.add_weighted_adapter`).

- **lora_activation**: Per-question time to set the newly merged adapter as the active one for inference (`model.set_adapter`).

- **tokenization** and **inference**: Per-question time for generation using a short prompt.

MULTI-LoRA (DYNAMIC)

This method loads the required LoRA modules from disk for each query, representing a scenario where preloading is not feasible.

- **model_loading**: One-time cost to load the base LLM.

- **rag_setup**: Same as the RAG method.

- **query_embedding** and **faiss_search**: Per-question retrieval time to identify relevant LoRAs.

- **lora_loading**: Per-question time to dynamically load the top-3 retrieved LoRA adapters from disk into memory. This represents a significant I/O overhead.

- **lora_merge** and **lora_attach**: Per-question time to merge the newly loaded adapters and set the result as active.

- **tokenization** and **inference**: Per-question generation time.

EXPERIMENTAL RESULTS

The results of the timing experiment, as visually detailed in the stacked bar chart in Figure 9, confirm several initial hypotheses while also revealing unexpected interactions between LoRA modules and underlying model optimizations. The figure provides a clear breakdown of total execution time into its constituent components for each method, which are analyzed below.

**Overall Performance Comparison.** As hypothesized, the In-Context Learning (ICL) method was by far the most time-consuming. Its total processing time was dominated by the 'inference' stage, a direct consequence of the substantial computational cost of applying self-attention over the full document context for every query. Among the LoRA-based methods, a key finding is the significant advantage of the preloading strategy. The Multi-LoRA (Preloaded) configuration, which loads all necessary adapters into GPU memory once at startup, was considerably faster than the Multi-LoRA (Dynamic) approach. The latter was bottlenecked by the per-query I/O latency of loading adapters from disk. This result underscores that while LoRA offers efficient inference, the surrounding architecture for managing and serving modules has significant potential for optimization.

**Inference Time and Flash Attention Interactions.** A deeper analysis of the 'inference' component, particularly concerning the use of Flash Attention, yielded more intriguing results. When experiments were run with Flash Attention disabled, the performance aligned with general expectations for context-heavy methods; ICL's inference time was overwhelmingly longer than all other methods. However, an unexpected discrepancy emerged between the closed-book methods. The inference time for the Single LoRA configuration was approximately twice as long as that of the Base Model, despite both processing identical short-context prompts (i.e., the question only). Theoretically, the raw computational cost for inference should be nearly identical, as the number of floating-point operations is not significantly increased by the LoRA adapter. This discrepancy strongly suggests that the presence of the PEFT adapter may inadvertently prevent the underlying model from activating certain default optimization strategies during its forward pass. Conversely, when experiments were run with Flash Attention enabled, another counter-intuitive trend was observed. The

inference speed of the Base Model paradoxically decreased compared to its non-Flash-Attention run, while the inference speed for the Single LoRA model significantly improved, becoming faster than the base model.

