# OpenReview forum: "Understanding LoRA As Knowledge Memory: An Empirical Analysis"
_ICLR.cc/2026/Conference — Submitted to ICLR 2026_

### Official Review · Reviewer_iruZ · 2025-10-27

**Soundness:** 3
**Presentation:** 3
**Contribution:** 3
**Rating:** 6
**Confidence:** 4

**Summary:**

The work presents the first systematic study of using LoRA as knowledge memory. Comprehensive exmperiments are conducted, with corresponding empirial analysis. This study reveals the potential of LoRA (single or multi-) to serve as a standalone incremental knowledge update module or a complementary module of RAG or ICL, once configured properly. Synthetic datasets are built to investigate the influences on information capacity, data diversity, and model scalability for single LoRA.
The findings on data-parameter efficiency motivates the authors to develop mult-LoRA systems, which show competitive performance with good routing or merging strategies on both synthetic and real-world datasets, although lower than ICL and RAG. Further experiments show ICL and RAG combined with LoRA knowledge modules can boost the performance, especially, ICL + multi-LoRA provides the most promising results. Thus, multi-LoRA provides a possibility to serve as a supplemantary module to ICL, so that the combination can effcientive equip an out-of-date model with up-to-date knowledge.

**Strengths:**

1. The idea is novel, which may opens a new door to solve the problem that LLMs lacking up-to-date knowledge.
2. As the initial trial of this idea, a good amount of experiments are conducted to study the major aspects of the problem, which helps illustrate the performance landscape under typical use cases.
3. Synthetic datasets are built in a way to minimize the possiblity of knowledge contamination.
4. Clear take away messages are delivered.

**Weaknesses:**

1. To further demonstrate the efficacy of the proposed method, experiments under more realistic settings are needed. For example, news of different categories, such as politics, sports, and finance, are updated incrementally every day. Before and after the LLM's training end date, the things going on are mostly related logically, while the datasets used in the experiments are mostly standalone and only in 1 special category and are not in the time-incremental fashion. Time dependency and topic diversity may change the landscape.
2. According to the analysis, multi-LoRA has many advantages over single LoRA, especially with external support. But reliable and high-performing merging or routing methods are not fully explored.
3. In Table 1, the results in one dataset for one LLM family are not sufficient to draw a general conclusion on which single method or combination of methods work the best. Experiments on more models and datasets should be provided.
4. Regardless of 3, ICL + mullti-LoRA seems the best performing combination. However, it also combines the disadvantages of both: ICL is slower than RAG, and multi-LoRA is slower than single.
5. The code is not yet released.

As the initial work of this idea, I can accept imperfect answers for 1 and 2. But sufficient justifications to 3 and 4 should be provided.

**Questions:**

1. "Threshold" is mentioned in Fig.2 and also in the appendix, but not formally defined. Please provide the definition and motivation of setting this threshold.
2. The lora ranks of PhoneBook experiment is provide in line 1272--1273. But this critical param is not provided in any other settings, especially not for the benefit in time section.
3. In fig.5's 1st panel, any reason ICL is much better than the single-LoRA?
4. In fig.5's 2nd panel, RAG (text embedding) is said to underperform single-LoRA, but its bar is higher.

---

> ### Author Response · Authors · 2025-11-22
>
> We are grateful to the reviewer for recognizing this work as the 'first systematic study' of LoRA as knowledge memory and for appreciating the novelty of our approach. We are particularly pleased that you found our experiments on synthetic datasets well-designed to minimize contamination and our key takeaways clearly delivered.We have provided detailed responses to each of your comments and questions in the sections below.
>
>
> > ### **W1: Evaluation on realistic settings with time-dependency and diverse topics.**
>
> > ### **W2: Exploration of reliable and powerful merging and routing mechanisms.**
>
> We agree that validating the method in realistic, time-dependent environments and employing sophisticated routing are crucial for practical deployment. However, as we mentioned in General Response, the primary objective of this paper is to analyze the **fundamental potential** of LoRA as a flexible LLM memory form. We view the implementation of a full-scale system for continuous updates, what might be termed a "Continual LoRA Memory" framework, as a substantial undertaking beyond the current scope.
>
> In such a realistic scenario, we envision constructing a cumulative **"LoRA library"** by training modules on distinct time units or categories. As this library scales, the robust retrieval and merging mechanisms mentioned by the reviewer become essential. In this initial study, we prioritized training efficiency and deliberately avoided complex routing overhead. Notably, in preliminary experiments (not included in the paper), we explored **summary-based routing** and **alternative PEFT merging techniques,** but they did not yield significant performance gains compared to our simpler baseline. Consequently, we identify the development of scalable retrieval algorithms for the LoRA library as the primary direction for future work.
>
> > ### **W3: Additional experiments on different model families and datasets are needed.**
>
> We agree that evaluating on additional models and datasets will strengthen our claims. We are **currently conducting additional experiments** on the Qwen model family and the QuALITY dataset, and we will report the detailed results before the discussion period ends.
>
> - **Model Expansion (Qwen 3):** We initially focused on Llama to strictly control for potential data contamination observed in other open-weights models. We are now rigorously re-evaluating the Qwen 3 family to ensure a fair comparison and will add these results to Table 1.
> - **Dataset Expansion (QuALITY):** To further validate our method's effectiveness in long-context scenarios, we are incorporating the QuALITY dataset. This benchmark focuses on reasoning over long input texts, which serves as a strong complement to NQA for assessing the model's memory capabilities.
>
> > ### **W4: Concerns regarding trade-off between the inference latency and performance.**
>
> We acknowledge that the combined approach incurs higher computational costs compared to RAG or single-adapter setups. **However, our primary goal is not to position LoRA-based memory as a universal SOTA replacement for existing methods, but to validate it as a viable "third option" in the memory design space.** As explicitly stated in our conclusion, "LoRA-based memory should not be viewed as a replacement for context-based methods like RAG or ICL." Instead, our work aims to characterize the trade-offs of this new parametric approach, offering it as a complementary alternative for scenarios where context-based retrieval is insufficient or inappropriate, even if it comes with specific latency considerations.
>
> > ### **W5: Availability of the source code.**
>
> We are committed to open-sourcing our implementation to ensure reproducibility. We are currently setting up an anonymous GitHub repository and will share the link in the comments within the discussion period.

---

> > ### Author Response · Authors · 2025-12-01
> > **Update on W3: Additional experiments on different model families and datasets are needed.**
> >
> > We have successfully completed the additional experiments on the Qwen model family and the QuALITY dataset, and the results are presented in the **[updated Table 1](https://github.com/anonymous-iclr2026-loramem/ICLR2026_LoRA-As-Knowledge-Memory/blob/main/REBUTTAL/table1_Qwen_QuALITY.png)**
> >
> > The results from the **Qwen models exhibit trends consistent with those observed in the Llama family**, reinforcing the generalizability of our proposed method across different architectures.
> >
> > In the QuALITY dataset, **the overall performance hierarchy is largely maintained (Multi-LoRA Top 3 > Single LoRA > Multi-LoRA Top 1)**. We observe that In-Context Learning (ICL) performs strongly in larger models like Llama-3.1-8B and Qwen3-8B because the articles in QuALITY are significantly shorter than those in NarrativeQA, allowing these capable models to process the context effectively without relying heavily on external memory mechanisms. However, smaller models such as Llama-3.2-1B and Qwen3-1.7B continue to demonstrate clear performance benefits from our LoRA-based approach, validating its **effectiveness particularly for models with limited capacity or context windows.**

---

> ### Author Response · Authors · 2025-11-22
>
> > ### **Q1: Clarification for the thresholds in Fig. 2.**
>
> The "threshold" refers to the specific target metric value used to evaluate performance: **Accuracy** for the PhoneBook dataset and **Efficacy** for the CounterFact dataset.
>
> Regarding the motivation, we selected these thresholds heuristically to visualize meaningful trends across varying ranks. Extreme values yield uninformative results: setting the threshold too low leads to performance saturation (where all ranks succeed), while setting it too high causes all configurations to fail. Therefore, we chose three representative values within an effective range to clearly demonstrate how performance correlates with rank changes without hitting ceiling or floor effects.
>
> > ### **Q2: Request for explicit specification of LoRA ranks across all experimental settings, particularly regarding the time efficiency analysis.**
>
> We appreciate you pointing out this omission. We have revised the manuscript to explicitly report the LoRA rank parameters used in all experimental settings in **Appendix K, M, N and Q.** These additions have been marked in red in the revised paper/appendix.
>
> > ### **Q3: Clarification on why ICL outperforms Single-LoRA in the first panel of Figure 5.**
>
> The performance gap stems from the limited capacity of a Single-LoRA to encapsulate the extensive amount of information required in this specific setting. The volume of target knowledge exceeds what a single LoRA module can effectively learn, causing the training to fail (underfitting) and resulting in performance lower than the ICL baseline. Crucially, this section demonstrates the advantage of our Multi-LoRA approach: **under the exact same parameter budget, Multi-LoRA successfully absorbs the information and significantly outperforms both Single-LoRA and ICL.**
>
> > ### **Q4: Discrepancy between the visual bar height and the reported performance of RAG in Figure 5.**
>
> We thank the reviewer for spotting this inconsistency. You are correct that the bar was plotted incorrectly; we have **corrected Figure 5** in the revised manuscript to accurately match our analysis, showing that RAG indeed underperforms single-LoRA.

---

> ### Author Response · Authors · 2025-11-23
> **Update on W5: Availability of the source code.**
>
> As promised, we are now sharing the link to our anonymized source code to ensure reproducibility:
>
> **https://github.com/anonymous-iclr2026-loramem/ICLR2026_LoRA-As-Knowledge-Memory**
>
> Please let us know if you have any issues accessing the repository.

---

### Official Review · Reviewer_ZQ6Q · 2025-10-31

**Soundness:** 3
**Presentation:** 3
**Contribution:** 2
**Rating:** 4
**Confidence:** 4

**Summary:**

The paper offers a systematic and comprehensive empirical study of Low-Rank Adaptation (LoRA) modules as "knowledge memory" for large language models (LLMs). Departing from conventional memory-augmentation approaches like RAG or ICL, the authors analyze LoRA’s scalability, finite capacity, parameter efficiency, optimal single-module usage, strategies for combining/merging multiple LoRAs, and interactions with external context for complex multi-hop scenarios. The work includes custom benchmarks (PhoneBook, PaperQA), extensive experiments on capacity scaling, data engineering, merging techniques, and computational efficiency, culminating in practical guidelines for deploying LoRA as a modular, efficient, and complementary memory solution alongside existing retrieval or in-context strategies.

**Strengths:**

- Experiments are thorough and well-controlled, including custom datasets (PhoneBook, PaperQA) designed specifically to measure symbolic memorization, generalization, and internalization of new knowledge. The methodology is transparent, and setup/hyperparameters are clearly explained.
- The manuscript is well-written and organized, making the takeaways easily accessible for future readers.

**Weaknesses:**

- The paper repeatedly claims to be “the first systematic” analysis of LoRA as knowledge memory, but prior work has already explored modular LoRAs, merging, and parametric memory (e.g., PRAG [1], DyRAG [2],  LoRA soups [3], LoRAHub [4]) in closely related scopes; the current framing doesn’t clearly delineate what is truly new beyond the chosen evaluation mix and efficiency metric.
- Evaluation relies heavily on synthetic or simplified setups. Two of the core testbeds—PhoneBook and the CF slices—measure keyed recall or local factual “edits” that may not reflect messy, open-domain knowledge integration or multi-document reasoning encountered in practice. The strict EM on PhoneBook and efficacy on small CF subsets may overstate real-world utility (Sec. 3; Fig. 1–2).
- LLM-as-judge (PaperQA) introduces circularity and bias. PaperQA’s scoring uses GPT-4.1 on a 0–10 scale; this risks format overfitting (since QA is also the training target) and inherits judge bias without human calibration or inter-rater checks (Sec. 4; PaperQA description). No agreement statistics, rubric validation, or robustness to prompt variance are provided.
- Base-model scaling experiment is under-controlled. All Qwen sizes are trained with identical hyperparameters and raw data (Fig. 4 left), which may be sub-optimal for some sizes and confound the non-monotonic pattern. Per-size LR/step tuning or compute-matched comparisons would be more informative.


[1] Parametric Retrieval Augmented Generation.

[2] Dynamic Parametric Retrieval Augmented Generation for Test-time Knowledge Enhancement

[3] LoRA Soups: Merging LoRAs for Practical Skill Composition Tasks

[4] LoraHub: Efficient Cross-Task Generalization via Dynamic LoRA Composition

**Questions:**

- Will more advanced LoRA techniques, such as DoRA and PiSSA, be more effective?
- Can the authors delineate precisely what’s new vs. prior multi-LoRA/merging/modular-memory work, and add a comparison table?
- In Fig. 1/7, the paper shows rank-dependent saturation. Could you provide per-layer/per-module breakdowns (e.g., attention vs. MLP adapters) to clarify where capacity limits arise?

---

> ### Author Response · Authors · 2025-11-22
>
> We sincerely thank the reviewer for the detailed evaluation and for recognizing our work as a 'systematic and comprehensive empirical study' with 'thorough and well-controlled' experiments. Please find our detailed responses to your questions and concerns in the subsequent sections.
>
> > ### **W1: Delineation of our paper from prior works (e.g., PRAG, DyPRAG, LoRA Soups, LoRAHub) is needed.**
>
> We thank the reviewer for the constructive feedback regarding the positioning of our work. We appreciate the opportunity to clarify our contribution.
>
> **1. Clarification on "First Systematic Analysis"**
> When we claim to provide the "first systematic analysis," we are not proposing a new memory method. Instead, our contribution lies in comprehensively examining the **intrinsic properties and feasibility** of using LoRA as a new memory option. Our work aims to provide the foundational empirical evidence, the "physics" of LoRA memory, that future system-level works can build upon.
>
> **2. Delineation from Prior Works (Summary of General Response)**
> As detailed in our **General Response**, we strictly delineate our work from the cited papers by addressing the specific gaps they leave.
>
> - **Vs. PRAG [1] & DyRAG [2]:** First, while these works primarily evaluate their systems on short-context retrieval (e.g., short Wikipedia snippets), our analysis shifts the focus to the challenge of memorizing **long contexts** directly related to specific queries. Second, they propose *systems* without analyzing the *memory unit's* limits. Our work fills this gap by empirically justifying the necessity of modular design (**Q2**), validating TIES over simpler methods like CAT (**Q10**), and identifying the optimal number of merges (**Q11**). We also define the essential data "recipe" (diversity and quality, **Q4, Q5**) that generation mechanisms like DyRAG require to be effective.
> - **Vs. LoRAHub [3] & LoRA Soups [4] :** These works focus on merging *procedural skills* (Task LoRAs) and rely on optimization-based merging (e.g., learning weights). In contrast, we specifically investigate *declarative knowledge* storage and strictly evaluate **training-free merging strategies** (Linear, Simple Concat, TIES). This distinction is crucial for our target "plug-and-play" scenario, where dynamically retrieving and swapping memory units must be done efficiently without the prohibitive computational overhead of learning merge coefficients for every combination.
>
> **3. Revisions in the Paper**
> We agree that these distinctions should be explicitly stated in the paper. We have revised the **Introduction** and **Related Works, and Appendix A (Related Works in Details)** sections to clearly delineate our contributions from these prior studies. The changes are highlighted in red in the revised manuscript.
>
> > ### **W2: Concerns regarding the reliance on synthetic or simplified testbeds for open-domain generalization.**
>
> We acknowledge that PhoneBook and CF slices are simplified; however, this design was **intentional to allow for a precise, granular analysis** of model behaviors rather than merely chasing top-line metrics. Specifically, these datasets provide a controlled environment free from pre-trained knowledge contamination, which is essential for accurately measuring scaling properties and retrieval mechanics.
>
> This methodological choice aligns with recent analytic studies in the field. For instance, **Allen-Zhu et al. [5]** employed synthetic biographies structurally similar to our PhoneBook to investigate knowledge storage, and **Frosio et al. [6]** utilized CounterFact to analyze how to retrieve context information. Furthermore, to address the concern regarding real-world applicability, we have also conducted and reported experiments on complex, open-domain benchmarks such as **PaperQA and NarrativeQA**, demonstrating that our method remains effective in messy, multi-document reasoning scenarios.

---

> ### Author Response · Authors · 2025-11-22
>
> > ### **W3: Concerns regarding the potential bias of the LLM-as-judge metric.**
>
>
> We acknowledge the concern that relying solely on LLM-based evaluation may introduce circularity or bias. To demonstrate the robustness of our results, we are incorporating the following additional validations:
>
> - **Objective Metrics:** We have measured performance using standard lexical metrics (ROUGE-L and BLEU). Our preliminary analysis shows that these scores exhibit trends consistent with the current results, indicating that the performance gains are not merely due to format overfitting. We will provide the detailed comparison table by next week.
> - **Human Calibration:** To validate the LLM scoring, we are currently conducting a human A/B test to check for alignment between human preference and the PaperQA scores. We will report these agreement statistics during the discussion phase. We also commit to expanding the sample size of human evaluators in the final version to ensure statistical significance and robust inter-rater reliability.
>
> > ### **W4: Fixed hyperparameter configuration of base-model scaling trends is under-controlled.**
>
> We acknowledge the reviewer's concern that using identical hyperparameters for all Qwen sizes might lead to sub-optimal performance for certain models and potentially confound the scaling analysis.
>
> To address this and ensure a rigorous comparison, we are **currently conducting a grid search** to identify the optimal hyperparameter configuration for each model size. We will share the results of these controlled experiments as soon as they are completed to verify the validity of the observed non-monotonic pattern.
>
> > ### **Q1: Advanced LoRA variants (e.g., DoRA, PiSSA) are more effective?**
>
> We agree that investigating the compatibility and potential performance gains of more recent PEFT methods is a valuable direction to demonstrate the generality of our approach. To address this, we are **currently running additional experiments** comparing the standard LoRA baseline with DoRA [7], and PiSSA [8]. We will provide the detailed results and analysis in a follow-up comment as soon as these experiments are completed.
>
> > ### **Q2: Request for clarification on the distinction between our work and prior multi-LoRA/modular memory studies.**
>
> We thank the reviewer for suggesting this crucial clarification. As requested, we have provided a detailed discussion on the distinction between our work and prior arts (Multi-LoRA, Merging, Modular-memory), along with a **comprehensive comparison table**, in our **General Response**.
> > ### **Q3: Request for granular breakdown (e.g., per-layer or per-module) of the rank-dependent saturation to identify specific capacity limits.**
>
> We agree that analyzing how saturation varies across different components (e.g., Attention vs. MLP) or layers is valuable for pinpointing the specific sources of capacity bottlenecks. We are currently conducting these additional experiments to provide the requested breakdown and will share the results as soon as they are available during the discussion period.
>
> [1] Parametric Retrieval Augmented Generation (Su et al., 2025)
>
> [2] Dynamic and Parametric Retrieval-Augmented Generation (Su et al., 2025)
>
> [3] LoraHub: Efficient Cross-Task Generalization via Dynamic LoRA Composition (Huang et al., 2023)
>
> [4] LoRA Soups: Merging LoRAs for Practical Skill Composition Tasks (Prabhakar et al., 2025)
>
> [5] Do LLMs dream of elephants (when told not to)? Latent concept association and associative memory in transformers (Frosio et al., 2024)
>
> [6] Physics of Language Models: Part 3.1, Knowledge Storage and Extraction (Allen-Zhu et al., 2023)
>
> [7] DoRA: Weight-Decomposed Low-Rank Adaptation (Liu et al., 2024)
>
> [8] PiSSA: Principal Singular Values and Singular Vectors Adaptation of Large Language Models Meng et al., 2024)

---

> ### Author Response · Authors · 2025-11-27
> **Update on W3: Concerns regarding the potential bias of the LLM-as-judge metric.**
>
> **Our additional experiments using objective metrics and cross-model validation support the robustness of LLM judge against circularity and bias.**
>
> **First, the reliability of our LLM-based evaluation is reinforced by its consistency with standard lexical metrics ($BLEU$ and $ROUGE-L$).** As illustrated in the linked figure **[[BLEU Figure](https://github.com/anonymous-iclr2026-loramem/ICLR2026_LoRA-As-Knowledge-Memory/blob/main/REBUTTAL/fig3_left_bleu.png), [ROUGE-L Figure](https://github.com/anonymous-iclr2026-loramem/ICLR2026_LoRA-As-Knowledge-Memory/blob/main/REBUTTAL/fig3_left_rougel.png)]**, the performance trends measured by these metrics closely align with the PaperQA scores. This correlation suggests that the observed improvements likely reflect genuine gains in information retention rather than artifacts of the judge's bias toward specific phrasing or reasoning styles.
>
> **Furthermore, our results with a different backbone model help alleviate concerns regarding potential format overfitting—the suspicion that the score favors QA-style outputs.** We replicated the experiment using Qwen-3 (**[Figure](https://github.com/anonymous-iclr2026-loramem/ICLR2026_LoRA-As-Knowledge-Memory/blob/main/REBUTTAL/qwen/fig3_left.png)**), and unlike Llama where QA pairs yielded the best performance, the Qwen-3 model achieved its highest scores with Summary-based synthetic data. If the evaluation system were primarily driven by a bias toward the QA format, we would expect QA data to consistently dominate regardless of the model architecture. Instead, the varying optimal formats across models suggest that the scoring mechanism is sensitive to content effectiveness rather than a rigid format preference.
>
> To further calibrate these findings, we are currently finalizing human preference testing and collecting agreement statistics between human annotators and the LLM judge. We look forward to sharing these results within the discussion period.

---

> ### Author Response · Authors · 2025-11-27
> **Update on Q1: Advanced LoRA variants (e.g., DoRA, PiSSA) are more effective?**
>
> We have completed the additional experiments, **which reveal that advanced PEFT techniques can significantly enhance memory performance.** As shown in the **[Figure](https://github.com/anonymous-iclr2026-loramem/ICLR2026_LoRA-As-Knowledge-Memory/blob/main/REBUTTAL/lora_variant_comparison.png),** while DoRA exhibits performance comparable to the standard LoRA baseline, **PiSSA demonstrates a remarkable improvement from 5.40 to 7.17** in PaperQA.

---

> ### Author Response · Authors · 2025-11-27
> **Update on Q3: Request for granular breakdown (e.g., per-layer or per-module) of the rank-dependent saturation to identify specific capacity limits.**
>
> As a follow-up to our previous response, we have completed the additional breakdown experiments **[Figure](https://github.com/anonymous-iclr2026-loramem/ICLR2026_LoRA-As-Knowledge-Memory/blob/main/REBUTTAL/phonebook_breakdown.png),** **which reveal distinct saturation trends across components and highlights potential improvements for parameter efficiency.** In experiments on the PhoneBook dataset where we isolated updates to Attention versus FFN modules and Early (layers 0-15) versus Late (layers 16-31) layers, we observed that FFN adapters and Early layers tend to sustain performance more robustly than their counterparts. These results imply that capacity limits are module-dependent, suggesting that we can optimize the storage-to-parameter ratio by strategically allocating the budget to these more robust components rather than distributing it uniformly.

---

> > ### Author Response · Authors · 2025-11-28
> > **Update on W4: Fixed hyperparameter configuration of base-model scaling trends is under-controlled.**
> >
> > We have confirmed that the non-monotonic scaling pattern persists even after optimizing hyperparameters for each model size, validating that our initial observation was not an artifact of fixed training settings (**[Figure](https://github.com/anonymous-iclr2026-loramem/ICLR2026_LoRA-As-Knowledge-Memory/blob/main/REBUTTAL/fig4_left_hp_gridsearch.png)**). To address the concern regarding under-controlled experiments, we conducted a comprehensive grid search sweeping over learning rates of $\{1e-5, 5e-5, 1e-4, 5e-4\}$ and training steps of $\{250, 500, 1000, 2000\}$ to identify the optimal configuration for each Qwen variant. While the per-size tuning yielded marginal performance improvements compared to the original identical settings, the overall performance trajectory and the non-monotonic trend across model sizes remained consistent with our reported results. This stability indicates that the scaling behavior we observed is a robust characteristic of the method rather than a confounding result of suboptimal hyperparameter choices.

---

> ### Author Response · Authors · 2025-12-01
> **[Final Update on W3] Human Evaluation Results: Validating Alignment with LLM Judge**
>
> Following our previous update, we have completed **a human evaluation showing strong alignment with LLM judge.**
>
> **Methodology:** We randomly sampled 10 instances from the test set and recruited 11 graduate students specializing in Machine Learning as annotators. For each instance, annotators were provided with the **Question** and the **Ground Truth (GT)** answer. They were then asked to rank the outputs from three representative settings—**Rewrite 4, Summary 8, and QA 40**—based on semantic similarity to the GT and overall answer quality.
>
> **Results:** The human evaluation results demonstrate a strong alignment with the LLM judge's assessment. As summarized in the table below, the average rankings derived from human annotators follow the exact same preference order as the LLM judge scores (**QA 40 > Summary 8 > Rewrite 4**).
>
> | | Rewrite 4 | Summary 8 | QA 40 |
> | :--- | :---: | :---: | :---: |
> | LLM Judge AVG Rank | 1.85 | 1.79 | **1.48** |
> | Human AVG Rank | 2.02 | 1.95 | **1.18** |
>
> **Future Work:** While the current sample size (11 evaluators, 10 instances) was constrained by the tight rebuttal timeline, the clear trend in these results provides strong evidence for the metric's reliability. We commit to expanding this study with a larger set of annotators and samples for the Camera Ready version to ensure further statistical robustness.

---

### Official Review · Reviewer_N1sD · 2025-11-01

**Soundness:** 3
**Presentation:** 2
**Contribution:** 3
**Rating:** 4
**Confidence:** 5

**Summary:**

This paper presents a systematic empirical study of Low‑Rank Adaptation (LoRA) framed as a parametric knowledge memory for large language models. The authors characterize single‑module properties (capacity, saturation, and parameter efficiency) across synthetic and factual benchmarks, and introduce two new evaluation sets (PhoneBook and PaperQA). They then explore data‑engineering strategies (QA, summaries, rewrites) to maximize information density, and analyze how base model scale and generator quality affect LoRA’s internalization. Moving to multi‑module designs, the work evaluates routing and merging trade‑offs, comparing oracle vs. embedding‑based routing and several merging schemes, and studies hybridization with ICL/RAG on long, multi‑hop narratives. The empirical narrative is careful and broad: it foregrounds capacity/efficiency curves, the impact of synthetic formats, and the system‑level costs of routing/merging, while demonstrating that hybrid parametric–non‑parametric systems often yield the most robust performance.

**Strengths:**

+ The empirical design is comprehensive and systematic: controlled rank–capacity sweeps, capacity saturation and efficiency analyses, ablations on synthetic data formats and generator quality, and careful comparisons of routing/merging strategies against oracles and standard baselines (ICL/RAG). The study uses multiple datasets spanning synthetic, counterfactual, and long‑form multi‑hop settings, supporting robust conclusions.


+ The paper is well structured, moving from single‑LoRA properties to multi‑LoRA systems and hybrids. Figures and appendices clearly present experimental setups, prompts, and hyperparameters, facilitating reproducibility and interpretability of results.


+ By deriving actionable design principles, e.g., efficiency‑optimal rank regimes, the value of data‑format curricula, and the role of sophisticated merging, this work provides practical guidance for building updatable, efficient memory systems. Its hybrid findings (LoRA + ICL/RAG) offer a pragmatic pathway for real‑world deployment.

**Weaknesses:**

- Since the main contribution of this work is a comprehensive analysis of LoRA as a parametric memory, the authors should explicitly explain the contribution of the related work that they have mentioned in the introduction section (L54-63) and explain their differences to those works and also the contributions they have provided (one by one) in the introduction section. For an empirical study it is important to clearly point out the differences to emphasize the originality and more importantly contributions of the study.

- The omission of several contemporary LoRA‐related memory works weakens the authors’ assertion that their study is the first systematic framework for LoRA‐based knowledge memory. While the authors cite some key early efforts  failing to acknowledge other closely related or contemporaneous papers makes it appear as though the authors are either unaware of or downplay existing analyses. This gap can undermine the perceived novelty of their contribution and lead readers to question how their findings build on or differ from prior work. Since this work is an intensive empirical study, it is crucial for the authors to have a strong and solid related works study. Following I have listed some of the missing works:

   * LoRA-Augmented Generation (LAG) for Knowledge-Intensive Tasks – Fleshman & Van Durme (2025).

  * How Much Knowledge Can You Pack into a LoRA Adapter without Harming LLM? – Pletenev et al. (Findings of NAACL 2025).

  * WISE: Rethinking the Knowledge Memory for Lifelong Model Editing of Large Language Models – Wang et al. (NeurIPS 2024).

  * LoRASculpt: Sculpting LoRA for Harmonizing General and Specialized Knowledge in Multimodal Large Language Models – Liang et al. (CVPR 2025).

  Acknowledging these works will clarify that while the paper is a significant and comprehensive empirical synthesis, the concepts of LoRA‐based knowledge modules, parametric RAG and capacity analysis already have preliminary treatments in the literature.

  * LoRA Without Regret – Schulman (2025). (I consider this one a concurrent work, just added it here to encourage the authors to consider it in their future studies)

 * The core capacity/efficiency sweeps (rank vs. capacity/efficiency, saturation curves) and much of the routing/merging analysis are conducted mainly on Llama‑3.1. These are the paper’s backbone “final” insights, but they are not fully replicated on a second family.

 * There is no study showing the effect of adding same number of parameters to the base model vs. adding the same number of parameters via singe or multi LoRA. This is important to study specially for the multi-LoRA scenario where routing among different LoRAs somehow mimics an MoE setup, hence, to have an apple-to-apple comparison the approximate same number of parameters should be considered.

 * In the multi LoRA experiments comparison to single LoRA it is not clear that what is the combined number of parameters in the multi LoRA setup vs. the single LoRA

 * No study on done on more general tasks and capabilities of LLMs (on general benchmarks)

 * The submission lacks experiments into how LoRA-based parametric memory, multi-LoRA, RAG, and ICL affect hallucination tendencies, the authors could incorporate explicit hallucination evaluations alongside their existing knowledge-recall and efficacy metrics.

**Questions:**

* L28: why do you have citation for right after the "Large Language Models" at the start of the sentence? Is it because of a LLM hallucination in injecting citations in the sentences? Please check this section out.

 * In Q3 it is not clear that how number of memorized tokens are calculated.

---

> ### Author Response · Authors · 2025-11-22
>
> We thank for the detailed assessment and for highlighting the strength of our work in deriving 'actionable design principles' through a controlled and broad empirical narrative. We are particularly encouraged that the systematic structure of our analysis, moving from single-LoRA properties to hybrid systems, was recognized for supporting robust conclusions. In the following sections, we provide point-by-point responses to the concerns raised in the Weaknesses and Questions sections.

---

> ### Author Response · Authors · 2025-11-22
>
> > ### **W1: Clarification on the distinction between our work and prior works (PnP, PRAG, and SEAL)**
>
> We agree that we missed clearly situating our work and explicitly differentiating it from prior studies, which are crucial for highlighting the originality and contribution of our empirical analysis. As detailed in our **General Response**, we have conducted a thorough comparison between our study and prior works. Specifically, we have clarified that:
>
> - **PnP (Caccia et al., 2025):** PnP focuses on a training objective (Deep Context Distillation) to optimize a *single* Knowledge LoRA. While PnP answers *"What is the best way to train a single module?"*, our work addresses fundamental questions such as *"What are the capacity limits?"* and *"What happens when we combine multiple modules?"* (Q8-Q11). Furthermore, we demonstrate that prioritizing data quality allows us to outperform PnP in single LoRA settings without the high computational overhead of their distillation framework (Table 1).
> - **SEAL (Zweiger et al., 2025):** SEAL proposes a meta-learning framework where an LLM generates its own synthetic data for fine-tuning. Their focus is on *finding* the best synthetic data using computationally expensive outer-loop training. In contrast, our work analyzes the *properties* of data (Q4, Q5) and focuses on efficient, plug-and-play memory LoRAs, explicitly excluding the costly iterative meta-learning processes required by SEAL.
> - **PRAG (Su et al., 2025):** PRAG presents a system-level proposal for retrieving and merging document-LoRAs. While they *assume* effective storage and merging, they do not deeply test the limits. Our work provides the **foundational analysis** that PRAG lacks. Specifically, we empirically validate why modular designs are necessary (Q2), demonstrate why simple merging fails (supporting their use of TIES) (Q10), and investigate the optimal number of merges (Q11), which was not explored in their study.
>
> Also, we have **revised the introduction and related works** to include a more detailed discussion of the related works mentioned.
>
> > ### **W2: The omission of several contemporary, closely related works.**
>
> In addition to the clarification regarding the works mentioned in the Introduction (discussed above), we sincerely thank the reviewer for pointing out these specific contemporary studies. We acknowledge that discussing these works is essential to solidly situate our empirical framework within the rapid advancements of the field.
>
> As detailed in our **General Response (Section: Comparison with Related Works)**, we have expanded our **Related Work** section to explicitly include and analyze these papers. Our detailed comparisons are as follows:
>
> - **LAG (Fleshman & Van Durme, 2025):** LAG targets the *retrieval* aspect ("which LoRA to pick?") for large LoRA libraries, effectively serving as a solution to the "Routing Bottleneck" we identify in Q9. Our work is complementary, as we focus on the intrinsic properties of the knowledge modules themselves. We study the fundamental capacity (Q1-Q5) and explore merging (Q10, Q11) as an alternative strategy to the routing approach used in LAG.
> - **How Much Knowledge Can You Pack... (Pletenev et al., 2025):** We find this work strongly supports our findings. The "harm" (catastrophic forgetting) they observe when packing "Unknown" facts aligns precisely with the capacity saturation we systematically measure in Q2. Furthermore, their mitigation strategy (mixing data) is an instance of the principle we analyze in Q5 (Benefits of Data Diversity). Our work extends this by providing a broader analysis covering Rank (Q1, Q3), Data Format (Q4), and Multi-LoRA systems (Q8-Q11).
> - **WISE (Wang et al., 2024):** WISE addresses the "impossible triangle" in continual editing using a "dual parametric memory" architecture. Key distinctions lie in: (1) Architecture: WISE copies full network layers, whereas we employ LoRA for parameter efficiency; and (2) Objective: WISE focuses on mitigating forgetting in sequential editing, while we focus on quantifying the static storage capacity and efficiency of the memory module itself.
> - **LoRASculpt (Liang et al., 2025):** This work addresses catastrophic forgetting in Multimodal LLMs (MLLMs) using pruning and regularization. Their focus on MLLMs and specific training techniques to prevent forgetting is orthogonal to our comprehensive analysis of knowledge capacity, data formatting, and merging strategies in standard LLMs.
>
> *(We also appreciate the reference to the concurrent work by Schulman (2025), which we have cited for future exploration.)*
>
> By incorporating these comparisons, we have strengthened the manuscript (red text in Related Works and Appendix A, Related Works in Detail section) to clearly demonstrate how our findings build upon and complement these state-of-the-art studies.

---

> ### Author Response · Authors · 2025-11-22
>
> > ### **W3: Try other models in addition to llama-3.1**
>
> To address the concern regarding empirical scope, we are **currently re-running the full set of core experiments on the Qwen family** and **will provide the replicated results in the updated manuscript** by next week.
>
> > ### **W4: Comparison against the base model fine-tuned with an equivalent number of parameters added by single or multi-LoRA.**
>
> We have carefully considered how to implement a baseline where the "same number of parameters are added to the base model." However, we faced difficulties in defining a fair comparison setting:
>
> - **Full Fine-tuning:** Expanding the base model and full fine-tuning is computationally prohibitive and infeasible due to the lack of access to the original pre-training corpus.
> - **Partial Training:** Adding parameters and training them on local contexts effectively converges to an Adapter/LoRA-like methodology. Furthermore, training a dense model on specific contexts often results in overfitting or catastrophic forgetting, making it an ineffective baseline for evaluating "memory" capacity.
>
> Given the constraints mentioned above, we would value your advice on a suitable baseline configuration. If you have a specific setting in mind, we would be happy to consider it for further validation.
>
>
>
> > ### **W5: The number of parameter comparison between single and multi-LoRA’s are not clear.**
>
> We appreciate the reviewer for pointing out that the comparison between single-LoRA and multi-LoRA experiments. While Q8 explicitly lists parameter counts, we acknowledge that this detail was omitted for the NQA dataset experiments in Q14. Unlike the fixed-set experiments, the NQA task involves a dynamic retrieval process. Consequently, the number of active LoRA modules varies for each data instance depending on the number of retrieved contexts.
>
> In our Multi-LoRA setup for NQA:
>
> - The model utilizes an average of **8.6 LoRA modules** per instance.
> - Although the rank of the Single LoRA baseline is **4 times larger** than the rank of individual modules in the Multi-LoRA setup, the total aggregate parameter count for Multi-LoRA is approximately **2 times larger** than the Single LoRA baseline.
>
> To verify whether the performance gain is merely due to the increased number of parameters, we are currently conducting an additional control experiment. In this setup, we scale up the Single LoRA rank to $r=32$, bringing its total parameters to a scale comparable to the Multi-LoRA setup. We will report the results of this experiment shortly.
>
> > ### **W6: Lack of validation on LLM general tasks or benchmarks.**
>
> We appreciate this comment and would like to **clarify our paper's core focus**.
>
> Our primary contribution is the exploration of **LoRA's viability as a novel external memory component** for Large Language Models. In developing and testing this new architectural capability, we aimed to isolate and test its unique function: **the storage and retrieval of dynamic, context-specific information**.
>
> Consequently, the **long-context memory retrieval scenario** (large context followed by a specific query related to the context) was selected because it is the **most direct and challenging test case for an external memory system**. This scenario effectively demonstrates the mechanism's ability to operate independently of the model's pre-trained knowledge.
>
> General LLM tasks (such as reasoning, instruction following, or common sense) typically rely on the model's vast pre-trained weights and complex internal mechanisms, rather than the dynamic context memory we propose. Direct application to these general benchmarks was **not expected to yield immediate performance gains** and would not have effectively isolated the memory mechanism's impact. We consider the application of this LoRA memory mechanism to enhance performance on broader, general LLM tasks a **critical and promising direction for future work**. We would greatly value any specific recommendations the reviewer might have regarding general tasks that are particularly suitable for evaluating this type of external memory system. We are eager to incorporate such insights to strengthen our evaluation.
>
> > ### **W7: Lack of analysis on hallucination.**
>
> We believe that in the context of Knowledge Injection, our performance metrics (e.g., Accuracy) implicitly reflect hallucination tendencies. Since any output deviating from the injected ground truth is counted as an error, these metrics effectively capture the model's failure to ground its response in the provided memory. That said, if you have a specific independent metric in mind suitable for this setting, we would be happy to consider it.

---

> ### Author Response · Authors · 2025-11-22
>
> > ### **Q1: Citation after the “Large Language Models” in the introduction is strange.**
>
> The citation for "Large Language Models" was not due to an accidental model hallucination. It was **intentionally included as a general citation** to refer to the substantial body of prior work and foundation that defines the field of Large Language Models (LLMs) used in our work. We have reviewed the section and agree that the placement might be structurally ambiguous. To improve clarity and flow, we removed the citation in the manuscript.
>
> > ### **Q2: how the number of memorized tokens are calculated in Q3 is not clear.**
> >
>
> We thank the reviewer for the question regarding the specific tokenization used in our experiments. We have updated Section Q3 of the manuscript to explicitly state that the **Llama 3 tokenizer** was used for all token count measurements, ensuring there is no ambiguity regarding the metric.

---

> ### Author Response · Authors · 2025-11-27
> **Update on W3: Try other models in addition to Llama-3.1**
>
> **We have successfully replicated our core experiments on the Qwen 3 8B model, confirming that the key trends and insights reported in our paper are robust and generalizable across different model families.** The only notable difference we observed was in the synthetic data analysis: unlike our previous observations, the Qwen model demonstrated superior performance with Summary data compared to QA data. We present the quantitative results corresponding to Table 1 in the tables below.
>
> ### **Closed Book**
>
> | **Method** | **Llama-3.2-1B** | **Llama-3.1-8B** | **Qwen3-1.7B** | **Qwen3-8B** |
> | --- | --- | --- | --- | --- |
> | Base model | 9.08 | 13.08 | 13.27 | 16.35 |
> | Single LoRA | **23.81** | **27.05** | **24.78** | **25.78** |
> | Multi-LoRA Top1 | 16.87 | 19.95 | 15.90 | 19.13 |
> | Multi-LoRA Top3 | 16.85 | 22.42 | 16.27 | 21.15 |
>
> ---
>
> ### **Open Book**
>
> | **Method** | **Llama-3.2-1B** | **Llama-3.1-8B** | **Qwen3-1.7B** | **Qwen3-8B** |
> | --- | --- | --- | --- | --- |
> | ICL | 24.52 | 33.81 | 27.34 | 35.22 |
> | RAG | 21.90 | 29.20 | 23.80 | 31.15 |
> | Single LoRA + ICL | *24.73* | 35.39 | 28.37 | 32.34 |
> | Single LoRA + RAG | 24.12 | 32.18 | 27.12 | 31.37 |
> | Multi-LoRA Top1 + ICL | 20.57 | 33.62 | 19.09 | 24.48 |
> | Multi-LoRA Top1 + RAG | 19.22 | 25.03 | 19.10 | 24.42 |
> | Multi-LoRA Top3 + ICL | **26.41** | **38.78** | *29.17* | **37.23** |
> | Multi-LoRA Top3 + RAG | 24.53 | *35.55* | **29.10** | *37.64* |
>
> We have included other results at the following URL:
> **https://github.com/anonymous-iclr2026-loramem/ICLR2026_LoRA-As-Knowledge-Memory/tree/main/REBUTTAL/qwen**

---

### Author Response · Authors · 2025-11-22
**General Response: Clarification Contributions and Delineation from Related Works**

We thank the reviewers for their insightful feedback regarding the positioning of our work and the relationship with existing literature. We recognize that the distinction between our *comprehensive analysis* and prior *methodological proposals* needs to be made more explicit.

While several recent works utilize LoRA for knowledge storage or editing, our work is **not** proposing a new training objective or a specific retrieval system. Instead, we provide the **first systematic empirical analysis** focusing on the **fundamental properties, limitations, and scalability** of LoRA, explicitly investigating its potential as a **new option for LLM memory**.

Below, we clarify our contribution by delineating our work from the suggested literature, categorized by their primary focus.

---

> ### Author Response · Authors · 2025-11-22
>
> ### **Advanced Training Objectives for Single Memory Modules**
>
> - **Training Plug-n-Play Knowledge Modules with Deep Context Distillation (PnP) (Caccia et al., 2025)**
>     - **Contribution:** A superior *training objective* (Deep Context Distillation, DCD) for a *single* Knowledge LoRA, which distills the behavior of an in-context teacher model .
>     - **Focus / Limitation:** This work relies on a complex distillation framework. It incurs significant overhead by requiring a teacher model to process the full context and aligning hidden states across all layers.
>     - **Our Delineation:** While PnP is exclusively focused on training a single module, our work analyzes the module's fundamental capacity (Rank vs. Data Size) and the critical problem of combining multiple modules (Q8-Q11). PnP answers, "What is the best way to train a single module?" whereas we answer, "What are the fundamental limits?" and "What happens when we combine them?". Furthermore, Table 1 demonstrates that prioritizing data quality allows us to outperform PnP in single LoRA settings without the computational overhead of distillation.
> - **Self-Adapting Language Models (SEAL) (Zweiger et al., 2025)**
>     - **Contribution:** A meta-learning *framework* where an LLM uses Reinforcement Learning to *learn to generate its own* optimal synthetic data ("Self-Edits") for finetuning (which can be a LoRA) .
>     - **Focus / Limitation:** While SEAL’s *Knowledge Incorporation* task shares similarities with our concept of LoRA-as-memory, their framework operates under significantly heavier constraints. Specifically, SEAL requires training a generator LLM in an outer loop to produce implications (e.g., inferences, logical consequences). This creates two major bottlenecks: (1) the reliance on paired training data consisting of Context ($C$) and Downstream Task ($\tau$), and (2) the high computational cost of the iterative outer-loop process involving repeated LoRA fine-tuning.
>     - **Our Delineation:** SEAL provides a *method to find* the best synthetic data. Our work *analyzes* the properties of that data (Q4, Q5). In contrast, we consider plug-and-play memory LoRAs. We explicitly exclude the computationally expensive process of training synthetic data generators or performing iterative meta-learning.

---

> > ### Author Response · Authors · 2025-11-22
> >
> > ### **Parametric RAG Systems**
> >
> > - **Parametric Retrieval Augmented Generation (PRAG) (Su et al., 2025)**
> >     - **Contribution:** A *system* for 'Parametric RAG' that pre-trains document-LoRAs and retrieves/merges them at inference .
> >     - **Focus / Limitation:** A system-level proposal evaluated on general QA benchmarks (e.g., 2WikiMultihopQA, HotpotQA, PopQA) by memorizing short Wikipedia pages. It assumed LoRA modules could store document knowledge and be effectively merged (using TIES). However, it did not test the limits of this storage (capacity), justify why TIES was necessary, explore optimal number of LoRAs to merge.
> >     - **Our Delineation:** We provide the foundational analysis for PRAG, shifting the focus to memorizing long contexts directly related to specific queries. Our Q2 analysis explains why PRAG's modular (per-document) design is necessary. Q10 shows that simpler methods like CAT fail for knowledge merging (supporting TIES). Q11 explicitly investigates the optimal number of LoRAs to merge, filling the gap in PRAG's exploration.
> > - **Dynamic and Parametric Retrieval-Augmented Generation (Su et al., 2025)**
> >     - **Contribution:** A *system* to solve PRAG's storage cost by *dynamically generating* LoRA modules from document content using a "Parameter Translator".
> >     - **Focus / Limitation:** primarily focuses on the *mechanism* of generation (the translator) rather than the quality of the result. This approach has structural drawbacks:
> >         1. **High Overhead:** Training the Parameter Translator incurs significant overhead as it requires extensive prior LoRA tuning.
> >         2. **Generalization & Flexibility:** The translator's output space corresponds to the LoRA parameter count (millions), which hampers generalization and necessitates expensive retraining whenever the target LoRA architecture changes.
> >         3. **Lack of Analysis:** Crucially, it does not analyze the intrinsic properties (capacity, efficiency) of the LoRAs it generates.
> >     - **Our Delineation:** While DyPRAG focuses on *how* to create the LoRA module (with the aforementioned heavy machinery), our work focuses on the *optimal design* of the module itself. Our findings on data formatting (Q4, Q5) provide the essential "recipe" (e.g., use diverse, high-quality QA data) that any generation system, including DyPRAG, should target to be effective.
> > - **LoRA-Augmented Generation (LAG) for Knowledge-Intensive Language Tasks (Fleshman & Van Durme, 2025)**
> >     - **Contribution:** Proposes a *data-free routing/retrieval* system for large LoRA libraries (1000+ modules) . It combines fast "Arrow" filtering with accurate "SpectR" reranking .
> >     - **Our Delineation:** While LAG manages both task and knowledge LoRAs based on Wikipedia-level entries, our work exclusively focuses on *knowledge LoRAs* capable of encoding longer contexts. LAG targets the retrieval aspect ("which LoRA to pick?"), serving as a solution to the Routing Bottleneck we identify in Q9. Our work is complementary, as we analyze the intrinsic properties of these long-context knowledge modules (Q1-Q5) and explore merging (Q10, Q11) as an alternative strategy to simple routing.

---

> ### Author Response · Authors · 2025-11-22
>
> ### **Learned Merging for Task LoRAs**
> - **LoraHub: Efficient Cross-Task Generalization via Dynamic LoRA Composition (Huang et al., 2023)**
>     - **Contribution:** A method for merging *Task LoRAs* to achieve *cross-task generalization* using gradient-free optimization on few-shot examples .
>     - **Focus / Limitation:** Like LoRA Soups, its focus is entirely on *Task LoRAs* and procedural skills, not factual (declarative) knowledge storage.
>     - **Our Delineation:** We distinguish our work in two key aspects. First, unlike LoraHub's focus on procedural task adaptation, we investigate the potential of LoRA as a memory unit for storing factual knowledge. Second, regarding the merging mechanism, LoraHub relies on optimizing combination weights (albeit gradient-free). In our plug-and-play scenario where multiple memory LoRAs are dynamically retrieved and swapped, optimizing coefficients for every combination introduces prohibitive computational overhead and complexity. Therefore, we exclude learning-based merging and instead strictly compare training-free merging strategies, specifically Linear Merging, simple Weight Concatenation (distinct from the method in LoRA Soups), and TIES, as detailed in our response to Q10.
> - **LoRA Soups: Merging LoRAs for Practical Skill Composition Tasks (Prabhakar et al., 2025)**
>     - **Contribution:** A method for merging *Task LoRAs* (skills, e.g., math + code) by learning a merge weight (CAT) on a small data subset .
>     - **Focus / Limitation:** Explicitly focused on *Task LoRAs* ('knowing how'). Its findings on *skill* composition do not necessarily apply to *knowledge* storage. It noted merging 3+ modules degraded performance but didn't analyze the cause from a knowledge perspective .
>     - **Our Delineation:** Similar to LoraHub, this method focuses on procedural skills and requires optimization (CAT) for merging. Our approach differs by targeting factual memory and employing training-free merging (Linear, Simple Concat, TIES) to enable efficient plug-and-play usage without the overhead of learning merge weights (see Q10).

---

> > ### Author Response · Authors · 2025-11-22
> >
> > ### **Continual Learning**
> >
> > - **WISE: Rethinking the Knowledge Memory for Lifelong Model Editing of Large Language Models (Wang et al., 2024)**
> >     - **Contribution:** WISE proposes a "dual parametric memory" architecture to solve the "impossible triangle" in continual editing. Specifically, it preserves original parameters as "Main Memory" and copies specific Feed-Forward Network (FFN) layers to serve as "Side Memory" for storing edits. By training a routing mechanism to decide which memory processes a given query, it protects pre-trained knowledge while accurately applying edits.
> >     - **Our Delineation:** There are two key distinctions. First, regarding the architecture, WISE relies on copying full network layers, whereas we employ LoRA to instantiate memory, focusing on parameter efficiency. Second, regarding the objective, WISE focuses on mitigating forgetting in a sequential editing setting, while our work focuses on quantifying the static capacity and efficiency of the memory module itself.
> > - **LoRASculpt: Sculpting LoRA for Harmonizing General and Specialized Knowledge in Multimodal Large Language Models (Liang et al., 2025)**
> >     - **Contribution:** Addresses catastrophic forgetting in *MLLMs* during LoRA finetuning . It identifies "harmful redundancy" and proposes *pruning* (sparse updates) and a "Conflict Mitigation Regularizer"  to harmonize general and specialized knowledge.
> >     - **Focus / Limitation:** The methods (pruning, regularization)  and domain (MLLMs) are orthogonal to our analysis of knowledge capacity, data formatting, and merging in LLMs.
> >     - **Our Delineation:** LoRASculpt focuses on *how to train* a LoRA for MLLMs to prevent forgetting, using pruning and regularization. Our work focuses on analyzing the potential of LoRA as a memory.
> > - **How Much Knowledge Can You Pack into a LoRA Adapter without Harming LLM? (Pletenev et al., 2025)**
> >     - **Contribution:** Investigates the *harm* (catastrophic forgetting) caused by packing new, "Unknown" facts into a LoRA adapter. It finds that mixing "Unknown" and "HighlyKnown" facts mitigates this harm.
> >     - **Our Delineation:** This paper strongly *supports* our findings. The "harm" they observe (Fig 1) is precisely the capacity saturation we systematically measure in Q2 . Their mitigation (mixing data) is an instance of the principle we analyze in Q5 (Benefits of Data Diversity) . Our work provides a much broader analysis by *also* investigating the role of Rank (Q1, Q3), Data Format (Q4), Multi-LoRA systems (Q8-Q11), and RAG/ICL hybrids (Q13).

---

> ### Author Response · Authors · 2025-11-22
>
> We provide the following **comparison table** to highlight the comprehensive scope of our analysis relative to PnP, PRAG and SEAL.
> | Feature                          | Ours | PnP | PRAG | SEAL |
> |:---------------------------------|:----:|:---:|:----:|:----:|
> | Storage Capacity Analysis        |  O   |  X  |  X   |  X   |
> | Efficiency Analysis              |  O   |  X  |  X   |  X   |
> | Training Data Analysis           |  ◎   |  O  |  O   |  O   |
> | Multi-Module Merging             |  O   |  X  |  △   |  X   |
> | Merging LoRA Count Analysis      |  O   |  X  |  X   |  X   |
> | Long Context Memorization        |  O   |  O  |  X   |  X   |
> | Experiment with External Context |  O   |  O  |  O   |  X   |
> | Time Analysis                    |  ◎   |  O  |  O   |  X   |
>
> *Link for the detailed table will be provided.*
>
> In summary, our contribution is the establishment of a comprehensive **Physics of LoRA Memory**, mapping the relationships between rank, data, capacity, and merging, which serves as a necessary foundation for the advanced methods and systems cited by the reviewers. We have updated our Introduction and Related Works section to reflect these delineations clearly.

---

> ### Author Response · Authors · 2025-11-23
>
> As promised in our General Response, we hereby provide the detailed comparison table via the anonymous link below.
>
> **https://github.com/anonymous-iclr2026-loramem/ICLR2026_LoRA-As-Knowledge-Memory/blob/main/REBUTTAL/Analysis%20comparison%20table.xlsx**

---

### Author Response · Authors · 2025-11-23
**General Response: Release of Anonymized Source Code**

We have released the source code in an anonymous repository to ensure the reproducibility of our experiments. You can access the implementation details and scripts at the following link:

https://github.com/anonymous-iclr2026-loramem/ICLR2026_LoRA-As-Knowledge-Memory

We hope this helps clarify the implementation details. Please let us know if you have any issues accessing the repository.

---

### Author Response · Authors · 2025-12-03
**General Response: Rebuttal Summary Table**

| Reviewer | Scores (Soundness / Pres. / Contrib. / Overall) | Novelty & Delineation | Generalizability (Models/Tasks) | Evaluation Rigor | Practicality / Code | Main updates in Rebuttal | Follow-up / Current stance |
| --- | --- | --- | --- | --- | --- | --- | --- |
| **N1sD** | 3 / 2 / 3, Overall: **4** | $\checkmark$ (vs. LAG, WISE.) | $\checkmark$ (Requested non-Llama models) | $\checkmark$ (Param count fairness) | - | **Replicated all core exp. on Qwen family**; Expanded Related Works; Clarified param counts vs. Multi-LoRA. | Acknowledged strength of empirical design, and main request on model generalizability is addressed via Qwen results. |
| **ZQ6Q** | 3 / 3 / 2, Overall: **4** | $\checkmark$ (vs. PRAG, LoRAHub) | $\checkmark$ (Synthetic dependency) | $\checkmark$ (Judge bias, Hyperparams) | $\checkmark$ (Adv. PEFT: DoRA/PiSSA) | **Human Eval** & **BLEU/ROUGE** added (validating LLM judge); **DoRA/PiSSA** exp. added; Grid search conducted. | Main concerns are directly addressed in rebuttal: judge-bias concern mitigated via Human Eval and objective metrics; DoRA/PiSSA experiments added in response to their question. |
| **iruZ** | 3 / 3 / 3, Overall: **6** | - | $\checkmark$ (More datasets/models) | $\checkmark$ (Threshold def.) | $\checkmark$ (Code release, Real-world) | **Released Source Code**; Added **Qwen family** & **QuALITY** benchmark; Corrected Fig 5. | Initial requests (code, models, datasets) all fulfilled; explicitly values the work as “first systematic study.” |

---

> ### Author Response · Authors · 2025-12-03
> **Reviewer Feedback and Rebuttal Overview**
>
> This comment summarizes the overall state of the reviews and how the shared concerns have been comprehensively addressed.
>
> **Core Positioning of Our Work**: Before addressing specific reviewer concerns, we wish to clarify the primary value proposition of this paper, which serves as the foundation for our rebuttal. **This work does not aim to propose a specific "SOTA system" to replace RAG or ICL.** Instead, our goal is to provide the **first systematic empirical analysis** that establishes the fundamental properties or "physics" of the **LoRA-as-Memory axis**. We aim to validate this parametric approach as a viable, complementary third option for LLM memory and provide actionable design guidelines (e.g., capacity saturation, data recipes, merging dynamics) for the community.
>
> Across the reviews, the substantive concerns clustered into four themes, which we have extensively addressed with new experiments:
>
> - **Novelty and Delineation from Prior Work (PRAG, SEAL, LAG):** Reviewers (**N1sD, ZQ6Q**) asked for a clearer distinction between our analysis and prior modular LoRA proposals. We addressed this by clarifying that while prior works propose specific *systems*, our work investigates the *intrinsic mechanics* enabling those systems. We have rewritten the *Introduction* and *Related Works* (highlighted in red) and added a detailed comparison in the comment to explicitly delineate our contributions.
> - **Generalizability (Models and Tasks):** Reviewers (**N1sD, ZQ6Q, iruZ**) noted our initial reliance on the Llama-3.1 family and synthetic datasets. In response, we successfully replicated our core experiments on the **Qwen model family** (confirming consistent scaling trends) and included the **QuALITY benchmark.**
> - **Evaluation Rigor (LLM-as-Judge Bias, Hyperparameters):** Reviewer **ZQ6Q** raised concerns about potential bias in LLM-based evaluation and hyperparameter controls. We rigorously addressed this by conducting a **Human Evaluation study** (showing strong alignment with the LLM judge), reporting objective metrics (**BLEU/ROUGE**), performing a hyperparameter grid search to ensure fairness, and adding experiments with advanced PEFT methods (**DoRA, PiSSA**).
> - **Additional Concerns and Clarifications:** We also ran module and layer wise ablations (Attention vs. FFN, early vs. late layers), which showed that FFN adapters and earlier layers carry memory more robustly under capacity stress (**ZQ6Q**), and we addressed remaining technical points by defining the thresholds used in our plots and explicitly reporting LoRA ranks/parameters (**iruZ,** **N1sD**), correcting Fig. 5 (**iruZ**), and releasing anonymized code for reproducibility (**iruZ**).
>
> **Putting this together:**
>
> - **Reviewer iruZ (Score: 6)** explicitly values the work as the **"first systematic study"** that opens a new direction for solving LLM knowledge updates, and their requests on **code release** and **additional models/datasets** were fulfilled via the anonymized repo and added **Qwen/QuALITY** results.
> - **Reviewers N1sD & ZQ6Q (Score: 4)** acknowledge the study as **comprehensive and systematic**, but initially read it more as a method proposal than an analysis paper.
>     - **N1sD**’s main concern was **clearer positioning vs. related work** and **non-Llama replication**; we addressed this with an **expanded Related Work + comparison table**, **Qwen-family replication**, and clarified single vs. multi-LoRA parameter counts.
>     - **ZQ6Q**’s main concern was **novelty framing and evaluation rigor** (synthetic reliance, LLM-as-judge, scaling controls); we addressed this with extended Related Work, added **Qwen/QuALITY** experiments, **BLEU/ROUGE** and **Human Eval**, plus hyperparameter search and PEFT/module-wise ablations.
>
> **Conclusion:**
> We have moved beyond the initial submission by adding substantial empirical evidence (Qwen, QuALITY, Human Eval, DoRA/PiSSA) that reinforces our claims. Given that the shared concerns regarding scope and rigor have been explicitly resolved, we believe the revised manuscript and our answers now firmly establishes a robust framework for understanding LoRA as memory.

---

### Meta-Review · Area_Chair_GDXz · 2026-01-07

**Summary:**

This paper presents a large empirical study framing LoRA adapters as a parametric “knowledge memory” for LLMs. The authors systematically characterize single-LoRA behavior (capacity, saturation, parameter efficiency) across synthetic and factual settings, introduce two evaluation sets (PhoneBook, PaperQA), and study how data formats (QA, summaries, rewrites) affect how well knowledge is internalized. They then extend to multi-LoRA systems, analyzing routing (oracle vs embedding-based) and merging schemes, and finally evaluate hybrids that combine LoRA memory with ICL/RAG on longer, multi-hop narratives. Overall the paper aims to distill practical design principles for building updatable memory systems using LoRA modules.

Pros
1. Comprehensive empirical coverage: controlled rank/capacity sweeps, saturation/efficiency analyses, format and generator-quality ablations, and multi-LoRA routing/merging studies.
2. Clear system framing: moves from single-module “physics” to multi-module tradeoffs and hybrid LoRA+ICL/RAG recipes, with many actionable observations.

Cons
1. Novelty is primarily empirical synthesis rather than a new method: several reviewers note that modular LoRAs, merging/routing, and parametric RAG ideas have closely related prior work; the paper’s “first systematic” claim needs careful, narrowly scoped wording and a crisp contribution table.
2. Evaluation remains somewhat narrow/controlled: core insights are driven by synthetic or simplified setups (PhoneBook/CounterFact slices) and an LLM-judge benchmark (PaperQA) that, even with added validation, still raises questions about robustness and real-world generalization.
3. Some important comparisons are still missing or only partially resolved: e.g., clearer apples-to-apples parameter accounting in multi-LoRA vs single-LoRA across all settings, and more direct baselines for “same parameter budget” alternatives (dense expansion / MoE-like comparisons), plus limited discussion of hallucination/faithfulness impacts beyond accuracy-like metrics.
4. Practicality tradeoffs remain: multi-LoRA + ICL can perform best but may compound latency/complexity; routing/merging quality is critical and not fully explored beyond the studied heuristics.

The submission is a strong, well-executed empirical study with useful design guidance for practitioners interested in LoRA as a memory mechanism, and the rebuttal meaningfully improves positioning, scope (Qwen replication), and evaluation credibility (human alignment for judge-based scoring). Overall I view this as a solid piece of work, but not quite strong enough in novelty and validation breadth to clear the acceptance bar.

**Reviewer Concerns:**

1. Related work / novelty positioning: Authors added and discussed several missing contemporaneous LoRA-memory papers (e.g., LAG, Pletenev et al., WISE, LoRASculpt, etc.) and clarified how their study differs from PnP/SEAL/PRAG and other modular/merging lines.
2. “Only on Llama” generalization: Core results were replicated on Qwen (multiple sizes), and they report that key trends largely hold.
3. Clarity bugs and missing details: They fixed the odd citation placement, clarified how “memorized tokens” are counted (tokenizer), defined the “threshold” used in figures, and added missing LoRA-rank reporting plus a plotting error correction.
4. LLM-as-judge bias concern (PaperQA): They added corroboration with lexical metrics (BLEU/ROUGE-L), cross-model differences (QA vs summary format dependence), and a small human evaluation showing agreement with judge rankings.
5. “Advanced PEFT variants” question: Added results suggesting PiSSA improves over vanilla LoRA (DoRA ~ similar).
6. Where capacity bottlenecks arise: Added a per-component/per-layer breakdown (attention vs FFN, early vs late) suggesting FFN/early layers are more robust.

**Reviewer Scores:**

N/A

---

### Decision · Program_Chairs · 2026-01-26

Reject